# TGF-β reduces DNA ds-break repair mechanisms to heighten genetic diversity and adaptability of CD44+/CD24− cancer cells

Debjani Pal[1,2], Anja Pertot[1], Nitin H Shirole[1,3], Zhan Yao[1], Naishitha Anaparthy[1,2], Tyler Garvin[1,4], Hilary Cox[1], Kenneth Chang[1], Fred Rollins[1,4], Jude Kendall[1], Leyla Edwards[5], Vijay A Singh[5], Gary C Stone[5], Michael C Schatz[1,4], James Hicks[1,2,4,6], Gregory J Hannon[1,4,7], Raffaella Sordella[1,3,4]*

[1]Cold Spring Harbor Laboratory, Cold Spring Harbor, United States; [2]Graduate Program in Molecular and Cellular Biology, Stony Brook University, Stony Brook, United States; [3]Graduate Program in Genetics, Stony Brook University, Stony Brook, United States; [4]Watson School of Biological Sciences, Cold Spring Harbor Laboratory, Cold Spring Harbor, United States; [5]Huntington Hospital, Northwell Health, Huntington, United States; [6]University of Southern California, Los Angeles, United States; [7]Cancer Research UK – Cambridge Institute, University of Cambridge, Cambridge, United Kingdom

**Abstract** Many lines of evidence have indicated that both genetic and non-genetic determinants can contribute to intra-tumor heterogeneity and influence cancer outcomes. Among the best described sub-population of cancer cells generated by non-genetic mechanisms are cells characterized by a CD44+/CD24− cell surface marker profile. Here, we report that human CD44+/CD24− cancer cells are genetically highly unstable because of intrinsic defects in their DNA-repair capabilities. In fact, in CD44+/CD24− cells, constitutive activation of the TGF-beta axis was both necessary and sufficient to reduce the expression of genes that are crucial in coordinating DNA damage repair mechanisms. Consequently, we observed that cancer cells that reside in a CD44+/CD24− state are characterized by increased accumulation of DNA copy number alterations, greater genetic diversity and improved adaptability to drug treatment. Together, these data suggest that the transition into a CD44+/CD24− cell state can promote intra-tumor genetic heterogeneity, spur tumor evolution and increase tumor fitness.

*For correspondence: sordella@cshl.edu

**Competing interests:** The authors declare that no competing interests exist.

## Introduction

Differences in the morphology and behavior of cancer cells within tumors were first noted by pathologists in the 1800s. However, advancements in genome-sequencing technologies and the possibility of analyzing the genome at single-cell resolution have affirmed that cancer within a single patient is a heterogeneous mixture of genetically distinct sub-clones that can arise from the accumulation of random mutations during tumor initiation, progression and response to drug treatments (*Navin et al., 2011*; *Greaves and Maley, 2012*; *Burrell et al., 2013*). Tumors have been considered the result of neo-Darwinian (*Zellmer and Zhang, 2014*) evolution processes within tissues. However, ample evidence has also indicated that non-genetic determinants, including epigenetic modifications (e.g. chromatin modifications or the activities of microRNAs and other noncoding RNAs) and the capability of cancer cells to transition between various cell states, contribute to the diversity of cell

**eLife digest** A single tumor can be made up of thousands of cancer cells that look and behave differently from one another. This observed diversity arises from changes in the DNA sequence of particular genes, changes in the activity of genes, and the ability of cells to transition between different states. As a result, individual cells within a tumor may react differently to certain anti-cancer drugs: most of the cells may be sensitive to the treatment and die, whereas some might be resistant and survive.

Previous studies on several types of human cancers – including breast, brain and lung cancers – have identified a group of cells called CD44+/CD24- cells that seem to be more aggressive and resistant to therapy than other cancer cells. However, it is currently not known exactly how these CD44+/CD24- cells influence a whole tumor's resistance to anti-cancer drugs.

Certain cancer cells in tumors are exposed to a signal molecule called TGF-$\beta$. CD44+/CD24- cells are unusual in that they are able to produce and release this signal molecule themselves. Pal et al. show that in CD44+/CD24- cells from a human lung cancer cell line, TGF-$\beta$ decreased the activity of genes responsible for accurately fixing breaks in the CD44+/CD24- cells' DNA. As a result, these cells made more mistakes than other lung cancer cells when repairing damaged DNA and consequently accumulated additional genetic mutations. Furthermore, tumors containing these cells were more likely to survive treatment with chemotherapy.

The findings of Pal et al. show that the CD44+/CD24- cells exposed to TGF-$\beta$ had a survival advantage because they were more genetically diverse and therefore better able to adapt to new drug treatments and other changes in their surroundings. Future experiments may explore how to specifically target and kill the CD44+/CD24- cells from tumors.

type and to the cell behaviors observed within tumors (*Almendro et al., 2013*; *Kreso and Dick, 2014*).

One of the best examples of a cell state transition that responds to epigenetic/stochastic events is the differential expression of the cell surface markers CD44 and CD24 (*Al-Hajj et al., 2003*; *Polyak and Weinberg, 2009*; *Yao et al., 2010*; *Korkaya et al., 2011*; *Brooks et al., 2015*). CD44 and CD24 are cell surface glycoproteins that are reportedly involved in cell-cell and cell-matrix interactions, as well as in the regulation of cancer cell growth, anchorage-independent proliferation and survival (*Alho and Underhill, 1989*; *Birch et al., 1991*; *Sy et al., 1991*; *Jain et al., 1996*; *Goodison et al., 1999*; *Smith et al., 2006*).

CD44+/CD24− cell populations are particularly salient in the field of oncology. It has been shown that CD44+/CD24− cancer cells are more adept at forming tumors, are more resistant to chemotherapy and tend to have greater metastatic potential (*Al-Hajj et al., 2003*; *Polyak and Weinberg, 2009*; *Yao et al., 2010*; *Korkaya et al., 2011*; *Brooks et al., 2015*). CD44+/CD24− cells were initially identified in breast tumors and breast-cancer-derived cell lines and have since been observed in the vast majority of tumors and tumor-derived cell lines, including non-small-cell lung carcinomas (NSCLCs), glioblastomas, neuroblastomas and leiomyosarcomas, as well as pancreatic, colon, prostate and ovarian tumors/cancers. Although CD44+/CD24− cells are the most abundant sub-population of cells in certain tumors, in many cases, they constitute only 3–5% of total tumor cells (*Al-Hajj et al., 2003*; *Polyak and Weinberg, 2009*; *Yao et al., 2010*; *Korkaya et al., 2011*; *Brooks et al., 2015*).

From a molecular standpoint, CD44+/CD24− cells are characterized by mesenchymal-like features including a decreased expression of E-cadherin and an increase in both vimentin and the master regulators of epithelial to mesenchymal transition (EMT): TGF-$\beta$, Snail and Zeb2. These cells also exhibit upregulation of stem-cell markers, such as Oct3/4, Nanog, Sox2, IL-6, C-Myc and BMI-1, and of anti-apoptotic proteins, such as Mcl1, Bcl2 and Bcl-XL (*Al-Hajj et al., 2003*; *Polyak and Weinberg, 2009*; *Yao et al., 2010*; *Korkaya et al., 2011*; *Brooks et al., 2015*).

In addition to the aforementioned characteristics, we found that CD44+/CD24− cells have intrinsic defects in their ability to repair DNA lesions. In particular, we observed that these defects could be ascribed to the activation of TGF-$\beta$-mediated signaling; we discovered that this signaling

pathway was both necessary and sufficient for decreasing the expression of genes such as *BLM*, *BRCA2*, *FANCF*, *NBN*, *PMS1*, *RAD50*, *RDM1*, *WRN*, *ATM* and *ATR*, which are essential in homology-directed repair (HDR) of DNA double-strand breaks (DSBs).

TGF-$\beta$ was first described in human platelets as a secreted protein that played a potential role in wound-healing responses. Since then, the TGF-beta$\beta$ signaling pathway has been the focus of a multitude of studies. Upon activation, TGF-beta$\beta$ can bind to the TGF-$\beta$ receptor II (T$\beta$RII), which in turn phosphorylates the TGF-$\beta$ receptor I (T$\beta$RI) (*Massagué et al., 2005*). From here, the activated T$\beta$RI can trigger a signaling cascade that propagates and amplifies the signal through the phosphorylation of intracellular downstream effectors such as SMAD2 and SMAD3, AKT and MAPK (*Massagué et al., 2005*). In the case of SMAD2/3, these effectors can form a complex with the SMAD4 protein that translocates the nucleus where, upon binding to specific DNA sequence motifs and transcriptional regulatory complexes, they can trans-activate target genes (*Massagué et al., 2005*).

The activation of the TGF-$\beta$ signaling axis impacts diverse and often contrasting cellular processes, such as cell proliferation, epithelial to mesenchymal differentiation, migration, apoptosis and ECM remodeling (*Massagué, 2008*; *Derynck and Miyazono, 2008*; *Oshimori and Fuchs, 2012*). The duality of TGF-$\beta$-mediated signaling is best exemplified by the role of TGF-$\beta$ in tumorigenesis. In fact, depending on the context, TGF-$\beta$ can either halt or promote tumorigenesis (*Massagué, 2008*). Furthermore, TGF-$\beta$ signaling has recently been shown to mediate resistance to targeted and conventional anticancer agents through the activation of pro-survival pathways (such as those involving Il-6, Bcl2 or Mcl-1) or of cell metabolism (*Pham et al., 2007*; *Franco et al., 2010*; *Yao et al., 2010*; *Oshimori et al., 2015*).

Interestingly, previous observations have already implied a role for TGF-$\beta$ in DNA repair (*Glick et al., 1996*; *Preobrazhenska et al., 2002*; *Kirshner et al., 2006*; *Wiegman et al., 2007*; *Liu et al., 2014*). While some previous studies align with our findings and demonstrate a possible role for TGF$\beta$ in reducing the expression and/or activity of certain genes involved in DNA repair, other works have challenged this view by showing that inhibition of TGF-$\beta$ signaling tends to attenuate DNA damage responses.

Owing to these contradictory observations, we utilized a multifaceted approach to investigate whether a decrease in the expression of genes involved in homology directed repair (HDR) by TGF-$\beta$ was sufficient to induce DNA DSBs and an increased accumulation of DNA copy number alterations (CNAs). In fact, because of the decrease in the efficiency of HDR, a switch to other break-repair mechanisms, including non-homologous end joining (NHEJ) and microhomology-mediated break-induced replication (MMBIR) (*Hastings et al., 2009*; *Fitzgerald et al., 2017*), could give rise to an increase in the accumulation of chromosomal translocation and CNAs in CD44+/CD24− cancer cells and cells exposed to TGF-$\beta$. Consistent with the observation that TGF-$\beta$ decreases the expression of *BLM*, *BRCA2*, *FANCF*, *NBN*, *PMS1*, *RAD50*, *RDM1*, *WRN*, *ATM* and *ATR*, we found that cells exposed to TGF-$\beta$ have a decreased capacity to repair DNA DSBs. In addition, we also observed that CD44+/CD24− cells from tumor and tumor-derived cell lines, as well as cells that have been exposed to TGF-$\beta$, had an increased clonal genetic diversity.

Central to current cancer research is the notion that the acquisition of genetic mutations in cancer cells generate complex populations of cells that are subject to Darwinian evolution (*Cahill et al., 1999*; *Almendro et al., 2013*; *Burrell et al., 2013*). In keeping with this principle, much like the branching evolution that Darwin described, the progeny of a cancer cell with increased fitness thrives in certain environments and gives rise to a dominant clonal population (*Cahill et al., 1999*; *Almendro et al., 2013*; *Burrell et al., 2013*). Consistently, we observed that the increased genetic diversity in the cancer cell population that was induced by transient exposure to TGF-$\beta$ enabled cancer cell populations to better respond to multiple-drug treatments when compared to TGF-$\beta$-naïve cells. This observation is particularly interesting because previous works have already shown that TGF-$\beta$ could induce drug resistance by activating anti-apoptotic pathways (*Yao et al., 2010*). Therefore, the fact that TGF-$\beta$ could induce drug resistance by accelerating cancer evolution suggests that exposure to TGF-$\beta$ could possibly provide a survival niche for cancer cells, allowing them to develop more stable genetic traits and increased malignancy.

In summary, our findings imply that the tumor microenvironment and gene regulatory networks could generate phenotypic diversity, including cells defined by a CD44+/CD24− state, enabling the development of novel, heritable genotypic profiles that could be selected for. Therefore, our

findings could unify the non-genetic (Lamarckian) and mutation-driven (Darwinian) mechanisms that have been used to explain tumor adaptability and drug resistance. This unified theory surpasses the conventional models of tumorigenesis and somatic evolution in moving beyond simple Darwinian schemes.

## Results

### A genome-wide shRNA screen identifies genes involved in replication or DNA damage repair as essential for the survival of CD44 +/CD24− cells

To gain a better understanding of the biology of CD44+/CD24− cells and to identify genes that are selectively required for their survival, we performed an RNA interference (RNAi)-based, forward genetic screen. As an initial paradigm to study CD44+/CD24− vulnerabilities, we used H1650 and H1650-M3 cells. The latter are derived from H1650 cells and harbor both phenotypic characteristics (i.e., epithelial to mesenchymal transition, increased metastatic capacity, motility, invasion and resistance to drug treatment) and molecular characteristics (i.e., increased surface expression of CD44 and decreased expression of CD24, constitutive activation of the TGF-$\beta$ axis, and autocrine secretion of IL-6) of CD44+/CD24− cells (*Figure 1—figure supplement 1A*) (*Yao et al., 2010*). H1650 and H1650-M3 cells were infected with a retroviral shRNA library that was previously constructed in the pSM1 vector (*Paddison et al., 2004*). This library consists of 28,000 sequence-verified shRNAs designed to target approximately 9,000 genes that have been implicated in tumorigenesis. Each shRNA is linked to a unique 60-nucleotide sequence (its DNA 'barcode') that can be used to monitor relative frequencies of individual shRNAs in complex populations over time. Infected cells were grown, after which the representation of individual shRNAs was determined using barcode microarray analysis. DNA was extracted, amplified and hybridized on custom-made microarrays. In the context of our drop-out screen, the decreased abundance of a particular shRNA in the pool indicates that that shRNA is targeting a gene that is essential for proliferation and/or survival. An outline of the experiment is shown in *Figure 1—figure supplement 1B*. We found that 135 shRNAs reached a p-value of less than 0.5 and had reduced representations of more than 1-fold in H1650-M3 cells compared with H1650 cells (*Figure 1A*). Among the top shRNAs selectively depleted in the CD44+/CD24− H1650-M3 cells, we identified shRNAs targeting IL-6 (*Figure 1A*). This finding is consistent with previous reports (both from our group and others) indicating that the IL-6 axis is required for the survival of cells in a CD44+/CD24− state (*Yao et al., 2010*; *Marotta et al., 2011*). Interestingly, we also observed a decreased representation of shRNA-targeted genes in DNA repair/replication pathways such as *BRCA1, ORC5L, RFC3, POLS, ERCC8* and *RPA2* (*Figure 1A*).

More specifically, BRCA1 is part of a multi-protein complex that repairs DNA when both strands are broken. Mutations in or downregulation of the expression of this gene is associated with predisposition to cancers (*Ford et al., 1994*; *Miki et al., 1994*; *Thompson et al., 1995*). *ORC5L* encodes one of the six subunits that form the origin recognition complex (ORC) that is essential for the initiation of DNA replication in eukaryotic cells. In addition, ORC5L has been shown to be involved in other processes such as transcriptional gene silencing and sister chromatid cohesion in *Saccharomyces cerevisiae* (*Suter et al., 2004*). ERCC8 is part of the nucleotide excision repair (NER) pathway, a complex system that eliminates a broad spectrum of structural DNA lesions, including ultraviolet-induced pyrimidine dimers, chemical adducts and DNA cross-links (*Reardon and Sancar, 2005*). RPA2 is one of the three components of a protein complex involved in DNA replication, DNA repair and recombination. Interestingly, RPA2 phosphorylation is observed after the exposure of cells to ionizing radiation (IR) and other DNA-damaging agents, which suggests that the modified RPA2 protein participates in the regulation of DNA repair and/or DNA replication after DNA damage (*Zou et al., 2006*). When DNA replicative polymerases stall at sites of DNA lesion, translesion synthesis (TLS) polymerases such as POLS are recruited. POLS is also reported to be involved in cohesion at the replication fork in *S. cerevisiae* (*Wang et al., 2000*; *Wang and Christman, 2001*). Replication factor C3 (RFC3) has been shown to be part of multiple protein complexes that have distinct functions. In conjunction with RFC1, RFC2, RFC4, and RFC5, it forms a heteropentameric protein complex that is required for the loading of PCNA onto DNA at template-primer

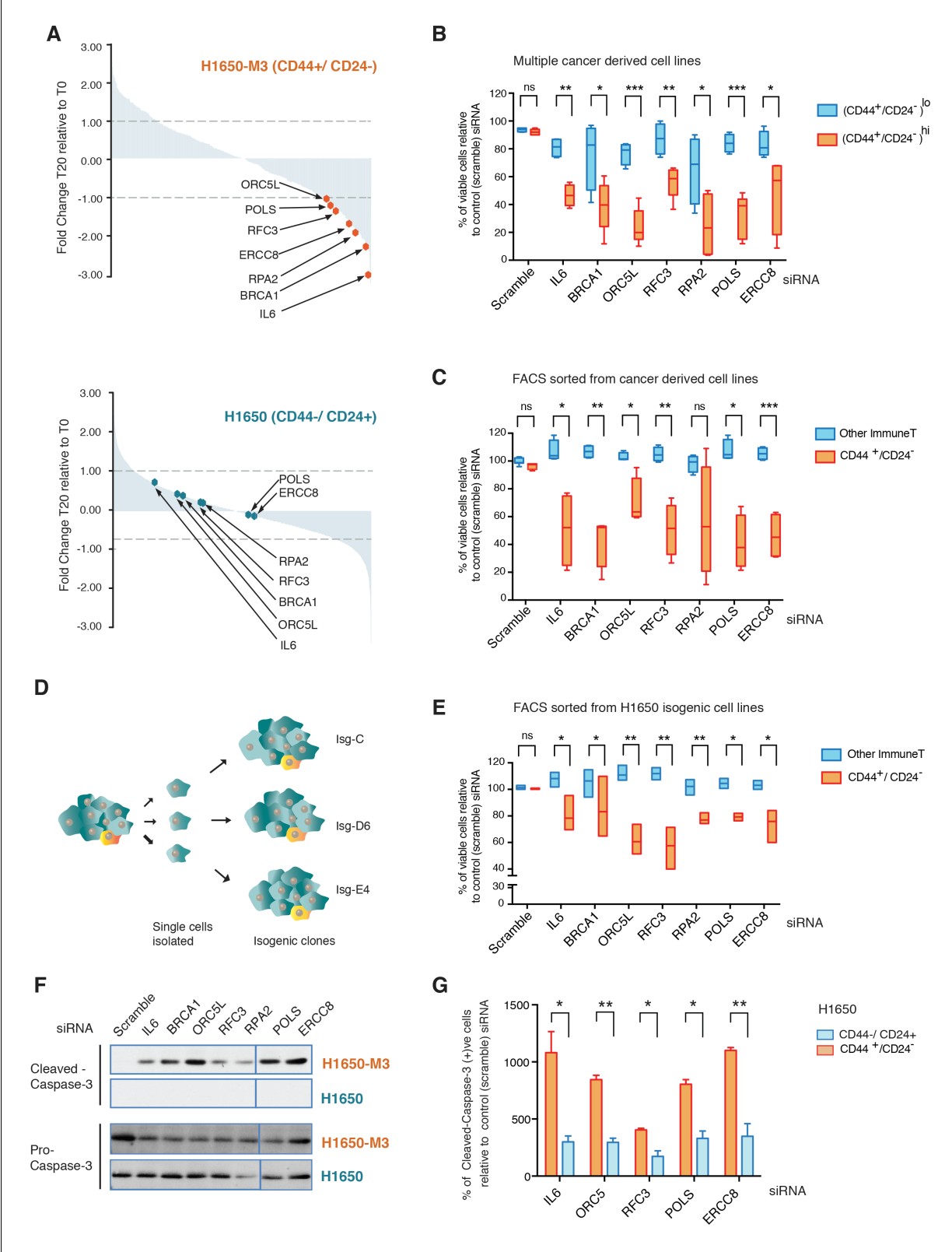

**Figure 1.** A genome-wide shRNA screen identifies genes involved in DNA damage repair (DDR) that are required for the survival of CD44+/CD24−. (**A**) The graph depicts the relative abundance of barcodes recovered from the screen. Each bar represents fold changes of an shRNA expression vector at T20 (i.e., 20 cell passages) compared with T0 (time of infection) in CD44+/CD24− H1650-M3 cells (upper panel) and CD44−/CD24+ H1650 cells (lower panel). Dots indicate unique genes, knockdown of which conferred proliferative disadvantage to CD44+/CD24− (H1650-M3) cells. The data are plotted

*Figure 1 continued on next page*

*Figure 1 continued*

as the means of three biological replicates in ascending order. A FACS profiling of H1650-M3 and H1650 cells, along with a schematic of the shRNA screen, is provided in *Figure 1—figure supplement 1*. (B) Validation of shRNA screen hits in tumor-derived cell lines characterized by low CD44+/CD24− cell content (i.e., MCF7, A549 and BT474) compared to cell lines with high CD44+/CD24− content (i.e., NCI-H23, PC9, MDA-MB435S and MDA-MB-231). The box plots show the percentage of viable cells 5 days after transfection with the indicated siRNAs relative to the number of control scramble-siRNA transfected cells. Each box is the mean ± SD of data collected from cell lines with either (CD44+/CD24−)$^{lo}$ or (CD44+/CD24−)$^{hi}$ content, from two independent experiments, each conducted in eight replicates (p-value *<0.05, **<0.005, ***<0.0005, unpaired t-test). FACS profiles for each cell line, relative % of viable cells for each cell line and knockdown efficiency are reported in *Figure 1—figure supplement 2*. (C) Validation of shRNA screen hits in tumor-derived cell lines FACS-sorted on the basis of the surface expression of CD44 and CD24. The box plots show the percentage of CD44+/CD24− cells and cells of other immune types upon transfection with the indicated siRNA oligonucleotides relative to control (scramble) siRNA. Each box is the mean ± SD of data collected from four cells lines (A549, H1650, PC9 and NCI-H23) upon FACS sorting, each from three replicates from two independent experiments. (p-value *<0.05, **<0.005, ***<0.0005, unpaired t-test). See *Figure 1—figure supplement 3* for more details. (D) Schematic of the generation of single cell-derived isogenic cell lines from H1650 cells. See *Figure 1—figure supplement 4A* for CD44 and CD24 surface marker staining profiles. (E) Validation of shRNA screen hits in the FACS-sorted H1650 single cell-derived isogenic clones—Isg-C, Isg-D6 and Isg-E4. The box plots indicate the percentage of CD44+/CD24− cells and cells of other immune types after transfection with the indicated siRNA oligonucleotides relative to control (scramble) siRNA. Each box is the mean ± SD of data collected from three different isogenic cell lines, each from three replicates from two independent experiments (p-value *<0.05, **<0.005, unpaired t-test). See *Figure 1—figure supplement 4B,C* for further details. (F) Expression of Pro-caspase three and Cleaved-caspase 3 (i.e., cell death marker) in H1650-M3 (CD44+/ CD24−) and H1650 (CD44−/CD24+) cell lines upon knockdown of indicated gene expression. Samples were collected 3 days post-transfection and protein lysates were immune-blotted with the indicated antibodies. Alpha-tubulin is used as the loading control. See *Figure 1—figure supplement 5* for quantification. (G) Percentage of Cleaved-caspase 3-positive cells, normalized to respective scramble controls (set at 100%), in FACS-sorted CD44+/CD24− and CD44−/CD44+ cells in H1650 cell line. Each bar represents mean ± SD of three replicates from two independent experiments(p-value *<0.05, **<0.005, unpaired t-test).

The following figure supplements are available for figure 1:

**Figure supplement 1.** Forward genetic screen performed in H1650 and H1650-M3 cell lines.

**Figure supplement 2.** Validation of shRNA screen hits in tumor-derived cell lines characterized by low (H1650, A549, MCF7, and BT-474) and high (PC9, NCI-H23, H1650-M3, MDA-MB-435s and MDA-MB-231) content of CD44+/CD24− cells.

**Figure supplement 3.** Validation of shRNA screen hits in tumor-derived cell lines FACS sorted on the basis of the surface expression of CD44 and CD24.

**Figure supplement 4.** Validation of shRNA screen hits in H1650-derived isogenic clones, FACS sorted on the basis of surface expression of CD44 and CD24.

**Figure supplement 5.** Quantification of apoptosis-mediated increase in lethality in H1650-M3 and H1650.

**Figure supplement 6.** Validation of shRNA screen hits in H1650 and H1650-M3 cells using the clonogenic assay technique.

junctions and for the polymerase switch between DNA polα and DNA polδ. As part of a complex that contains the RAD17 subunit, it regulates DNA damage checkpoints; in a complex with the CTF18 subunit, it is necessary for sister chromatid cohesion; whereas in a complex with ATAD5, it aids fork stalling recovery and DNA DSB repair (*Mayer et al., 2001*).

The identification of these genes in our drop-out screen was of particular interest as it indicated possible phenotype/cell state dependencies. As a first step to validating these findings, independent siRNAs were used to silence the expression of *IL-6, BRCA1, ORC5L, RFC3, POLS, ERCC8* and *RPA2* in H1650-M3 and H1650 cells, as well as in seven additional tumor-derived cell lines characterized by low (A549, MCF7 and BT474) or high (NCI-H23, PC9, MDA-MB435S and MDA-MB-231) content of CD44+/CD24− cells (*Figure 1B* and *Figure 1—figure supplement 2*). Overall, we observed that tumor-derived cell lines that had a high content of CD44+/CD24− cells were more sensitive to the inactivation of these genes.

Many studies have shown that stochastic, non-genetic processes can drive the acquisition of phenotypic differences among cancer cells (*Gupta et al., 2011*). This phenomenon is also evident in CD44+/CD24− cells. As shown by *Gupta et al. (2011)*, cancer cells grown in a uniform tissue culture *in vitro* microenvironment, when separated on the basis of the CD44 and CD24 cell surface markers, return to their original equilibrium proportion over a relatively short period of time.

Therefore, to determine whether the selective vulnerabilities that we identified in our screen also typified stochastically generated CD44+/CD24− cells, we extended our analysis to four pairs of tumor-derived (CD44+/CD24−)$^{lo}$ cell lines that were FACS-sorted on the basis of their surface expression of CD44 and CD24 (*Figure 1C* and *Figure 1—figure supplement 3*). Overall, as illustrated in *Figure 1C*, we observed that the knockdown of the genes we identified in our original drop-out screen resulted in higher lethality in cells that reside in a CD44+/CD24− state compared to cells of other immune types.

Cancer cells harbor many genetic alterations that, while important for tumorigenesis, can render them vulnerable to the loss of function of only one additional gene. To exclude the possibility that the selective vulnerabilities we observed in CD44+/CD24− cells were not due to a concealed genetic mutation but instead to an intrinsic property of the CD44+/CD24− state, we generated multiple single cell-derived cell lines from the H1650 cell line (*Figure 1D*). Within these isogenic cell lines, we then compared, at early passages, the effect of *IL-6*, *BRCA1*, *ORC5L*, *RFC3*, *POLS*, *ERCC8* and *RPA2* knockdown in FACS-sorted CD44+/CD24− cells and in cells of other immune types. Consistent with our previous data, we observed a decrease in the representation of CD44+/CD24− cells in the H1650 isogenic cell lines (H1650-Isg-C, -Isg-D6 and -Isg-E4) upon silencing of the indicated genes (*Figure 1E* and *Figure 1—figure supplement 4*).

The decrease in the number of CD44+/CD24− cells was not due to inter-conversion of cell states or a decrease in cell proliferation; rather, it was the result of increased lethality in the CD44+/CD24− cell populations upon silencing of the indicated genes. In fact, upon inactivation of *IL-6*, *BRCA1*, *ORC5L*, *RFC3*, *POLS*, *ERCC8* and *RPA2*, we observed (i) an increase in the expression of the pro-apoptotic marker Cleaved caspase-3 by western blot analysis in H1650-M3 cells compared to parental H1650 cells (*Figure 1F*, *Figure 1—figure supplement 5*); (ii) a higher number of Cleaved caspase-3 positive cells by FACS analysis in H1650 CD44+/CD24− cells compared to CD44−/CD24+ cells (*Figure 1G*) and (iii) a decrease in the number of colonies in the CD44+/CD24− cells in a standard clonogenic assay (*Figure 1—figure supplement 6*).

## CD44+/CD24− cells are characterized by decreased expression of homology-directed repair genes

Mutation analysis of *BRCA1*, *ORC5L*, *RFC3*, *POLS*, *ERCC8* and *RPA2* indicated that none of these genes were mutated in any of the cell lines utilized in our study (*Supplementary file 1*). Moreover, we neither observed any significant differences in the levels of their mRNA expression among the cell lines utilized (*Figure 2A and B* and *Figure 2—figure supplement 1*) nor detected significant variations in the efficiency of their knockdown (*Figure 1—figure supplement 2C*).

There is increasing evidence to suggest that mutations or the decreased expression of certain genome-stabilizing genes can weaken the intrinsic robustness of DNA repair mechanisms in tumor cells. Consequently, these cancer cells are rendered dependent on DNA damage/repair components distinct from the originally compromised gene (*Morandell and Yaffe, 2012*). Therefore, as a possible mechanism to explain the results of our functional genomic screen, we explored synthetic lethal interactions between gene products that have been reported to participate in DNA repair and those identified in our screen.

To this end, we generated gene expression profiles of H1650 and H1650-M3 cells and examined the relative abundance of DNA damage/repair pathway components (see *Supplementary file 2*), We did not observe major differences in the expression of genes involved in base excision repair, nucleotide excision repair, mismatch repair, or non-homologous end joining repair (*Supplementary file 2*). Yet, genes known to participate in HDR and DNA damage check-point regulation, such as *BLM*, *BRCA2*, *FANCF*, *NBN*, *PMS1*, *RAD50*, *RDM1*, *WRN*, *ATM* and *ATR* (*Ciccia and Elledge, 2010*), were expressed to a lesser extent in the CD44+/CD24− H1650-M3 cells than in the parental H1650 cells (*Supplementary file 2*).

We confirmed the observed differences in expression by analyzing these HDR genes by RT-qPCR in the H1650 and H1650-M3 cells (*Figure 2C* and *Figure 2—figure supplement 2*). To determine whether this result was a common feature of cells in a CD44+/CD24− state, we then measured the expression of *BLM*, *BRCA2*, *FANCF*, *NBN*, *PMS1*, *RAD50*, *RDM1*, *WRN*, *ATM* and *ATR* (i) in the tumor-derived cell lines utilized in our functional genomic screen (*Figure 2C* and *Figure 2—figure supplement 2*), (ii) in FACS-sorted H1650, A549 and MCF7 cells (*Figure 2D* and *Figure 2—figure supplement 3A*) and (iii) in four FACS-sorted primary human NSCLC tumors (*Figure 2E* and

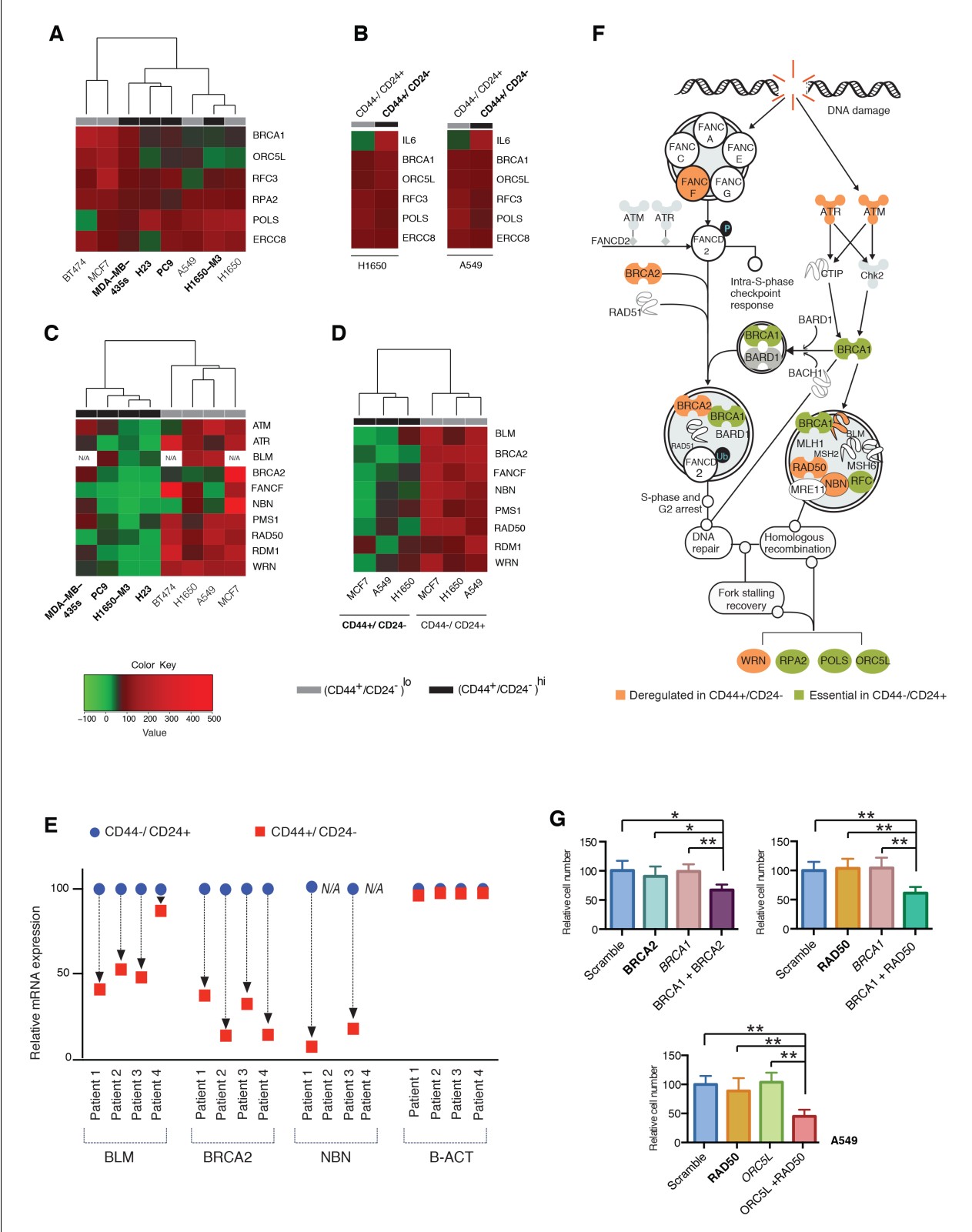

**Figure 2.** Decreased expression of homology-directed repair (HDR) genes in CD44+/CD24− cells results in synthetic lethal interactions. (**A**) The heat map represents a hierarchical cluster analysis of *BRCA1, ORC5L, RFC3, RPA2, POLS* and *ERCC8* mRNA expression in the indicated tumor-derived cell lines and (**B**) in H1650 and A549 cells that were FACS sorted on the basis of their surface expression of CD44 and CD24. mRNA expression was quantified by SYBR-green-based RT-qPCR. Cell lines with high CD44+/CD24− cell content are indicated in bold. The data represent mean ± SD of
*Figure 2 continued on next page*

*Figure 2 continued*

three replicates from two independent experiments. See *Figure 2—figure supplement 1* for details. (C) Hierarchical cluster analysis of the mRNA expression of the indicated HDR genes in multiple tumor-derived cell lines with high (indicated in bold) or low content of CD44+/CD24− cells. mRNA expression was quantified by SYBR-green-based RT-qPCR. The data represent the mean ± SD of three replicates from two independent experiments. See *Figure 2—figure supplement 2* for details. (D) Clustering analysis of mRNA expression of the HDR genes in CD44+/CD24− and CD44−/CD24+ cells FACS-sorted from H1650, A549 and MCF7 cells. mRNA expression was quantified by SYBR-green-based RT-qPCR. The data represent the mean ± SD of three replicates from two independent experiments. See *Figure 2—figure supplement 3A* for details. (E) Expression of *BLM*, *BRCA2* and *NBN* genes in FACS-sorted CD44−/CD24+ and CD44+/CD24− cells from four human primary NSCLC tumors. mRNA expression was quantified by SYBR-green-based RT-qPCR. Expression of an indicated mRNA in the CD44+/CD24− cells was calculated relative to its expression in CD44−/CD24+ cells from the respective tumor. Each dot represents mean ± SD of three replicates. See *Figure 2—figure supplement 3B,C* for details. (F) Schematic representation of the functional interactions between genes that we identified in the screen (green) and the HDR genes that we found to be downregulated (orange) in the H1650-M3 (and CD44+/CD24−) cells (see *Supplementary file 2* for details). (G) The charts depict the percentage of viable cells 5 days after knockdown of the indicated genes relative to a scramble-siRNA control in the A549 cell line. Each bar represents mean ± SD of eight replicates from two independent experiments. See *Figure 2—figure supplement 5* for knockdown efficiency (p-value *<0.05, **<0.005 paired t-test).

The following figure supplements are available for figure 2:

**Figure supplement 1.** mRNA expression analysis of the shRNA screen hits in tumor-derived cell lines and cells that have been FACS sorted on the basis of their surface expression of CD44 and CD24.

**Figure supplement 2.** Differential mRNA expression of homology-directed repair (HDR) genes in multiple tumor-derived cell lines.

**Figure supplement 3.** mRNA expression analysis of HDR genes in CD44−/CD24+ and CD44+/CD24− cells FACS sorted from cells lines and patient tumors.

**Figure supplement 4.** Differential expression of HDR genes in multiple tumor-derived cell lines at protein level.

**Figure supplement 5.** Knockdown efficiencies of the indicated siRNAs in A549 cells.

*Figure 2—figure supplement 3C*). In all cases, we observed that cells lines with (CD44+/CD24−)^hi content and FACS-sorted CD44+/CD24− cells exhibited decreased expression of these HDR genes.

Western blot analysis of BRCA2, RDM1 and WRN also indicated that the differential mRNA expression levels we observed were reflected by decreased protein expression (*Figure 2—figure supplement 4*). This observation was particularly interesting because many of the HDR genes that we detected as being downregulated in H1650-M3 cells have already been described as functionally interacting with the genes that we initially identified in our screen (*Figure 2F*) (*Ciccia and Elledge, 2010*).

To provide experimental evidence in support of the synthetic lethal interactions between the genes identified in the shRNA screen and the genes that had decreased expression in CD44+/CD24− cells, we partially knocked down expression of *BRCA2* and *RAD50* in A549 cells and then tested the viability of the cells upon combined silencing with the *BRCA1* or *ORC5L* genes (essential in CD44+/CD24− cells) (*Figure 2G* and *Figure 2—figure supplement 5*). We reasoned that decreasing the mRNA levels of *BRCA2* and *RAD50* would mimic their low expression in CD44+/CD24− cells. As a result, we found that the combined knockdown of *BRCA2* and *BRCA1*, *RAD50* and *BRCA1*, or *RAD50* and *ORC5L* resulted in a significant decrease in cell viability, even though the individual inactivation of *ORC5L*, *BRCA1*, *BRCA2* or *RAD50* had no significant effect on the viability of A549 cells (*Figure 2G*).

## TGF-β-mediated signaling controls the expression of the HDR genes that are downregulated in CD44+/CD24− cells

Cells in a CD44+/CD24− state are characterized by constitutive activation of the TGF-β axis that is both necessary and sufficient for the maintenance and/or acquisition of many of the features that characterize the CD44+/CD24− cell state (*Figure 3A and B* and *Figure 3—figure supplement 1*) (*Mani et al., 2008*). As previous studies have also indicated that TGF-β can modify the activity and expression of certain genes involved in DNA repair (*Glick et al., 1996*; *Harris et al., 1997*;

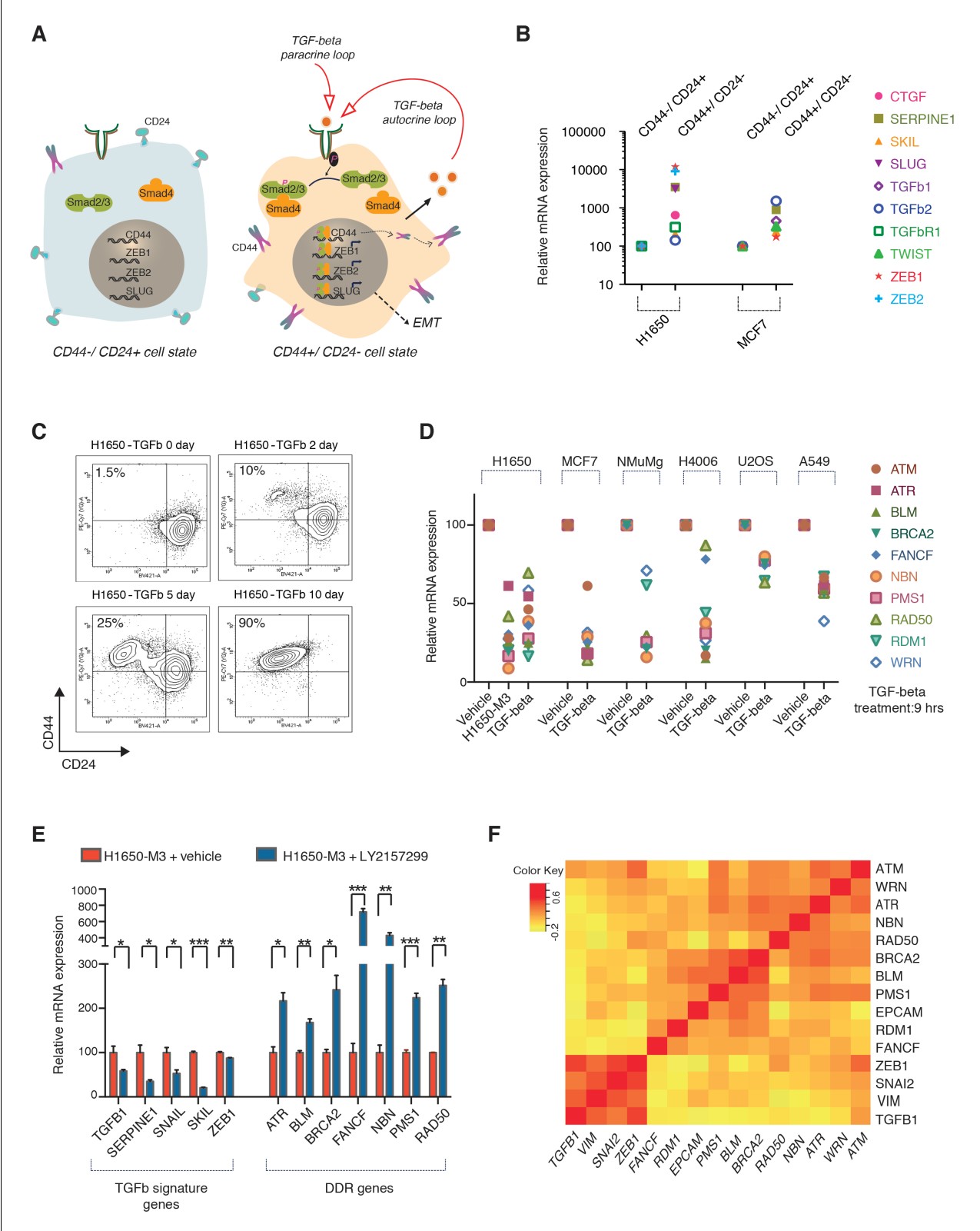

**Figure 3.** The TGF-$\beta$ axis controls the expression of the HDR genes that are downregulated in CD44+/CD24− cells. (**A**) TGF-$\beta$ signaling is required for CD44+/CD24− cell state transition. The schematic provided here is based on current literature (**Korkaya et al., 2011**; **Mani et al., 2008**). (**B**) mRNA expression analysis of well-known TGF-$\beta$ target genes in CD44+/CD24− cells relative to CD44−/ CD24+ cells FACS-sorted from H1650 and MCF7 cell lines. mRNA expression was quantified by SYBR-green-based RT-qPCR. Each dot represents the mean ± SD of three replicates from two independent

*Figure 3 continued on next page*

*Figure 3 continued*

experiments. See *Figure 3—figure supplement 1* for details. (**C**) FACS analysis of H1650 cells exposed to TGF-$\beta$. Cells were treated with TGF-$\beta$ for the indicated days and stained for the surface expression of CD44 and CD24 and analyzed by FACS. (**D**) mRNA expression analysis of the indicated HDR genes in TGF-$\beta$-treated cells relative to vehicle control across multiple tumor-derived cell lines. Cells were treated for 9 hr with TGF-$\beta$1 and TGF-$\beta$2 (1 ng/ml each). mRNA expression was quantified by SYBR-green-based RT-qPCR. Each dot represents the mean ± SD of three replicates from two independent experiments. See *Figure 3—figure supplement 2B* and *Figure 3—figure supplement 3* for additional details. (**E**) The inhibition of TGF-$\beta$ signaling in the CD44+/CD24− H1650-M3 cells results in an increased expression of HDR genes. TGF-$\beta$ receptor 1 (TGFBR1) kinase activity was blocked by treatment with 20 µM of LY2157299 (Selleckchem) for 72–96 hr. Expression of TGF-$\beta$ signature genes were used as a control for the efficacy of LY2157299 treatment. mRNA expression was quantified through SYBR-green-based RT-qPCR. Each bar represents the mean ± SD of three replicates from three independent experiments (p-value *<0.05, **<0.005 paired t-test). (**F**) Meta-analysis of human breast tumor dataset (BRCA) generated by the TCGA Research Network: http://cancergenome.nih.gov/. Average Pearson correlation coefficients (PCCs) between every pair of genes displayed were calculated and the matrix was generated. Pearson correlation coefficients ranged from −1 to +1. The matrix indicates an inverse co-regulation of the TGF-$\beta$1 and the HDR genes that we found to be downregulated in CD44+/CD24− cells.

The following figure supplements are available for figure 3:

**Figure supplement 1.** TGF-$\beta$ signaling is upregulated in FACS-sorted CD44+/CD24− cells.

**Figure supplement 2.** Active TGF-$\beta$ signaling control the expression of multiple HDR genes.

**Figure supplement 3.** TGF-$\beta$ signaling controls the expression of multiple HDR genes at the protein level.

**Figure supplement 4.** Inhibition of TGF-$\beta$ signaling in the CD44+/CD24− H1650-M3 cells results in an increased expression of HDR genes.

**Figure supplement 5.** Exposure to TGF-$\beta$ does not alter the cell cycle distribution in cancer cell lines.

*Preobrazhenska et al., 2002*; *Kirshner et al., 2006*; *Wiegman et al., 2007*), we examined whether TGF-$\beta$ could be responsible for the lower expression of the HDR genes that we observed in CD44+/CD24− cells.

By exposing cells to exogenous TGF-$\beta$, we confirmed that the TGF-$\beta$ axis was sufficient not only to induce the acquisition of mesenchymal-like features (*Figure 3—figure supplement 2A*) and the transition into the CD44+/CD24− state (*Figure 3C*) but also to decrease the expression of *BLM*, *BRCA2*, *FANCF*, *NBN*, *PMS1*, *RAD50*, *RDM1*, *WRN*, *ATM* and ATR in multiple tumor-derived cell lines (*Figure 3D* and *Figure 3—figure supplement 2B*). In the case of BRCA2, BLM and RAD50, we confirmed that TGF-$\beta$ also induced a decrease in protein expression as assessed through western blot analyses of cell extracts (*Figure 3—figure supplement 3*).

Conversely, treatment of the CD44+/CD24− H1650-M3 cells with two selective inhibitors of TGFBRI (LY2157299 and LY364947) augmented the expression of *ATR, BLM, BRCA2, FANCF, NBN, PMS1* and *RAD50* and, as expected, decreased the expression of the TGF-$\beta$-induced genes *SERPINE1, SNAIL, SKIL* and *ZEB1* (*Figure 3E* and *Figure 3—figure supplement 4*).

Altogether, these data indicate that TGF-$\beta$-mediated signaling is sufficient and necessary to decrease the expression of the HDR genes that we observed to be downregulated in CD44+/CD24− cells.

Next, to assess the relevance of this observation in human tumors, we mined a human breast tumor dataset (BRCA) generated by the TCGA Research Network (http://cancergenome.nih.gov/) (*Figure 3F*). We found that, consistent with our previous observations, both *TGFB1* and the TGF-$\beta$ signature genes were inversely correlated with the expression of *BLM, BRCA2, FANCF, NBN, PMS1, RAD50, RDM1, WRN, ATM* and *ATR* (*Figure 3F*).

It has been reported that TGF-$\beta$ can induce cell cycle arrest. Yet, in H1650 and U2OS cells, although TGF-$\beta$ reduced the expression of DNA repair genes within 9 hr (*Figure 3D*), it did not induce any significant changes in the distribution of cells in G1/S/G2 within the same timeframe (*Figure 3—figure supplement 5*). This observation strongly excludes the possibility that our findings could simply be explained by cell cycle alterations.

## TGF-β is sufficient and required in CD44+/CD24− cells to attenuate homology-directed repair mechanisms

From a functional standpoint, a decrease in the expression of HDR genes could result in an increased accumulation of DNA DSB (*Kwei et al., 2010*). To test whether this was the case in CD44+/CD24− cells, we compared γ-H2AX foci formation in FACS-sorted CD44+/CD24− and CD44−/CD24+ cells from H1650. We found that CD44+/CD24− cells had a higher number of γ-H2AX positive foci (*Figure 4A and B*). By staining the cells with 53BP1, a scaffold protein for DSB- responsive factors, we next assess if the increase in these foci was a consequence of a defect in DNA repair or DNA shredding (*Panier and Boulton, 2014*). In accordance with intrinsic defects in DNA damage/repair, we observed an increased occurrence of 53BP1/γ-H2AX double positive foci in CD44+/CD24− cells (*Figure 4A and B*) .

As an independent approach for evaluating DNA strand breaks, we also performed a comet assay (*Figure 4E*). The larger mean comet-tail movement in H1650-M3 cells compared with H1650 cells confirmed a higher number of DSBs in CD44+/CD24− cells (*Figure 4F and G*).

As we could show that TGF-β was both necessary and sufficient to regulate HDR genes, we next extended our analysis to A549, H1650 and MCF7 cells treated with TGF-β. Also in this case, we found that TGF-β treatment was sufficient to increase the number of 53BP1/γ-H2AX double-positive foci and the mean comet tail movement (*Figure 4C, D, F and G* and *Figure 4—figure supplement 1*).

In principle, an increased number of DNA breaks could be explained either by a decreased capability of DNA repair or by increased DNA damage. Hence, we used a homologous recombination reporter system (the DR-GFP reporter system) to further investigate the possibility that TGF-β exposure could result in defects in DNA repair (*Figure 4H*) (*Pierce et al., 1999*; *Nakanishi et al., 2005*; *Gunn and Stark, 2012*). This system is based on a non-crossover gene-conversion mechanism of two mutated GFP genes: (i) the SceGFP that is disrupted by an 18 bp recognition site for the I-SceI endonuclease and (ii) the iGFP that is truncated at the 5′ and 3′ ends. Upon transfection with pCBASce-I (expressing I-SceI), the SceGFP is cleaved and, because of an HDR event, a functional GFP gene (detectable by flow cytometry) is generated through gene conversion with iGFP. By using this assay, upon transfection with pCBASce-I, we observed a significant reduction in GFP-positive cells among U2OS cells as well as among H1650 cells treated with TGF-β (*Figure 4I* and *Figure 4—figure supplement 2*). As a control, we knocked down expression of *BRCA2* and *RAD50*, two well-known HDR genes (*Figure 4I* and *Figure 4—figure supplement 3*).

## The CD44+/CD24− cells are characterized by increased genetic diversity

Homology-directed repair deficiency is reported to induce a switch to non-homologous break repair mechanisms, such as NHEJ or MMBIR, which can lead to the accumulation of DNA 'joint points' where the non-homologous repairs occur(*Daley et al., 2005*; *Hastings et al., 2009*). These events have the potential to lead to chromosomal translocation, allelic imbalance and a specific DNA copy number profile named 'saw-tooth' (*Kwei et al., 2010*).

Whole-genome CNA analysis in FACS-sorted CD44+/CD24− and CD44−/CD24+ cells from three human NSCLC tumors and a tumor-derived cell line showed that the CD44+/CD24− cells possessed a higher content of DNA joint points and the distinctive saw-tooth profile (*Kwei et al., 2010*) (*Figure 5A, B, C and F* and *Figure 5—figure supplement 1A*). In addition, CNA analysis of a H1650 isogenic cell line (H1650-Isg-E4) also established that this was due neither to concealed genetic mutations (*Figure 5D, E and F* and *Figure 5—figure supplement 1B*) nor to the expansion of a single clonal population with high DNA DSBs (*Figure 5G* and *Figure 5—figure supplement 1B*), but rather to an intrinsic property of CD44+/CD24− cells.

On the basis of these observations, we concluded that the decreased expression of HDR genes and a reduced DNA repair proficiency could result in a hyper-mutable phenotype and an increased genetic diversity of CD44+/CD24− cells.

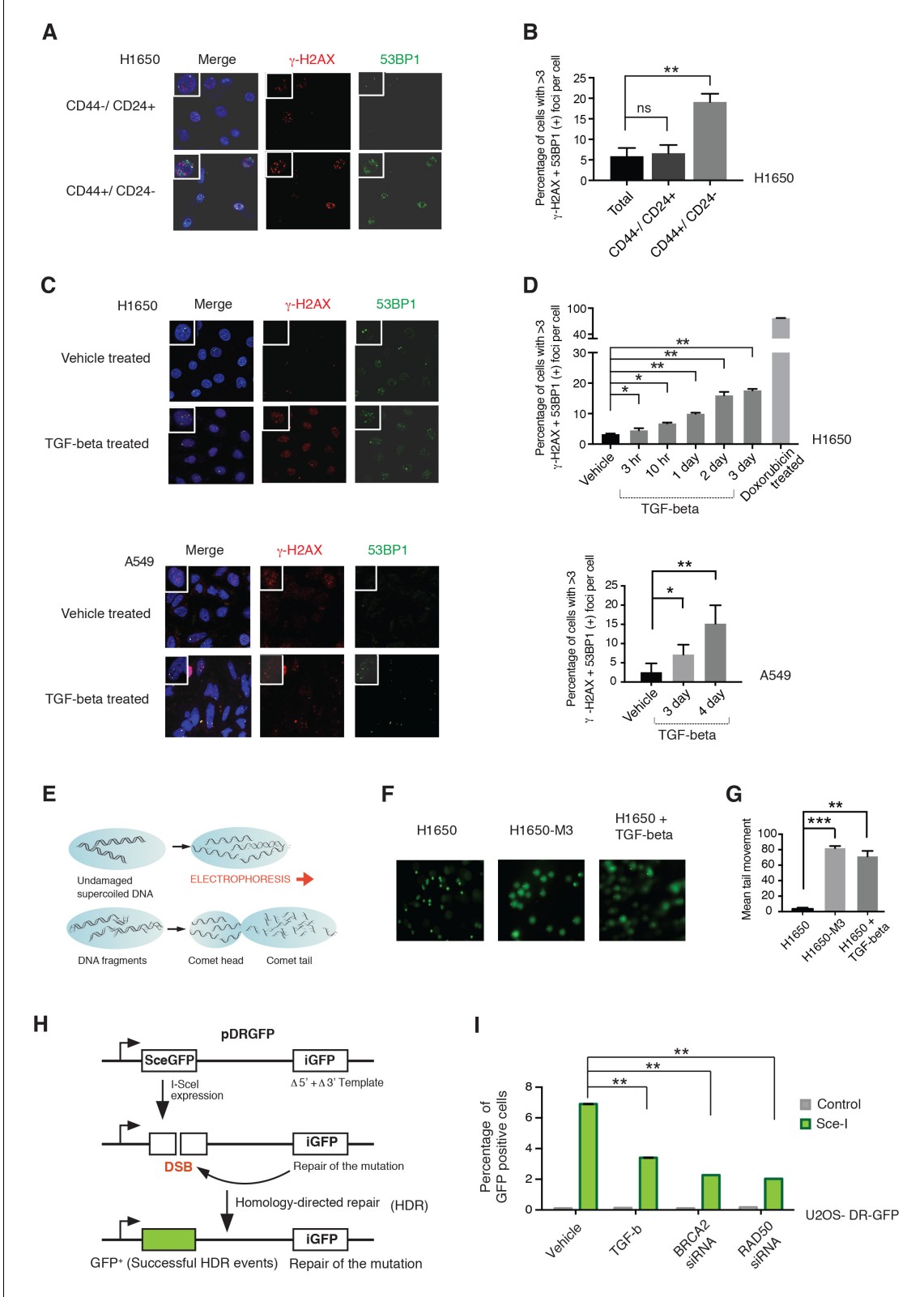

**Figure 4.** CD44+/CD24− cells are characterized by defect in DNA damage repair. (**A**) CD44+/CD24− cells are characterized by increased DNA DSBs compared with CD44−/CD24+ cells. CD44+/CD24− and CD44−/CD24+ cells sorted from H1650 were stained with antibodies against γ-H2AX (red) and 53BP1 (green). DAPI (blue) was used as a counter-stain. Insets in the left upper corner show a representative nucleus. (**B**) The chart represents quantification of the experiment depicted in (A). Each bar represents the mean ± SD of the percentage of cells with more than three γ-H2AX and 53BP1

*Figure 4 continued on next page*

*Figure 4 continued*

double-positive foci per field in CD44−/CD24+ cells and CD44+/CD24− cells, FACS-sorted from the H1650 cell line. Approximately ten fields were counted, for a total of 100 cells (n = 100) (p-value **<0.005, unpaired t-test). (C) H1650 and A549 cells treated with vehicle (DMSO) or TGF-$\beta$ (1 ng/ml of each of TGF-$\beta$1 and -$\beta$2, for 4 days) were stained with antibodies against γ-H2AX (red) and 53BP1 (green). DAPI (blue) was used as a counter-stain. Insets in the left upper corner show a representative nucleus staining; analysis of an additional cell line is provided in *Figure 4—figure supplement 1*. (D) The chart represents quantification of the experiment depicted in (C). Each bar represents the mean ± SD of the percentage of cells with more than three γ-H2AX and 53BP1 double-positive foci per field in vehicle or TGF-$\beta$ treated cells. Approximately ten fields were counted, for a total of 100 cells (n = 100). (p-value *<0.05, **<0.005, paired t-test). Doxorubicin treatment (10 µM for 24 hr) was used as a positive control. (E) Schematic of Comet assay. (F) Comet assay in H1650, H1650-M3 (CD44+/CD24−) and TGF-$\beta$-treated H1650 cells (treatment for 5 days), showing increased DNA strand breaks. (G) The chart represents quantification of the mean ± SD of tail movement of the samples depicted in (F), conducted in three replicates in two independent experiments. (p-value **<0.005, ***<0.0005, unpaired t-test). (H) Schematic of the DR-GFP assay. (I) The chart indicates the percentage of GFP-positive cells upon transfection with pCBASce-I (expressing Sce-I endonuclease) compared with control cells (untransfected cells). siRNA-mediated knock-down of RAD50 and BRCA2 was used as a homologous recombination (HR) efficiency control. Each bar represents mean ± SD of three replicates from two independent experiments (n = 6) (p-value **<0.0005, paired t-test). See *Figure 4—figure supplement 3* for knockdown efficiencies with the indicated siRNAs.

The following figure supplements are available for figure 4:

**Figure supplement 1.** TGF-$\beta$ treatment increases the number of 53BP1/γ-H2AX double-positive foci in MCF7 cells.

**Figure supplement 2.** TGF-$\beta$ reduces the efficiency of homologous recombination in a DR-GFP assay in H1650 cells.

**Figure supplement 3.** Knockdown efficiencies of the indicated siRNAs in U2OS-DR-GFP cells.

## Exposure to TGF-β is sufficient to increase both the CNA and the genetic diversity of the cell population

We showed that TGF-$\beta$ was both necessary and sufficient to reduce the expression of HDR genes and to increase the number of DNA DSBs in CD44+/CD24− cells. To assess whether TGF-$\beta$ treatment was also sufficient to increase the number of CNAs and the genetic diversity of cancer cells, we exposed H1650 isogenic cells (H1650-Isg-D6) to TGF-$\beta$ for six weeks and then performed whole-genome CNA analysis. As genetic alterations are stable, to exclude a possible direct effect of TGF-$\beta$-mediated signaling in our analysis of CNA, we performed our studies upon TGF-$\beta$ removal. Much like our observations in CD44+/CD24− cells, we detected an overall expansion in the number of DNA joint points and an increased genetic heterogeneity of H1650-Isg-D6 cells upon exposure to TGF-$\beta$ (*Figure 6A, B and C*).

## TGF-β-induced CD44+/CD24− cell state increases the adaptability of cancer cell populations

Since the conception of evolutionary reasoning, it has become evident that the stability and robustness of ecosystems depend on their diversity. In line with this principle, *Luria and Delbrück (1943)* proposed that the genetic diversity of a bacterial population was responsible for the resistance of the population to bacteriophage infection (*Luria and Delbrück, 1943*). In keeping with this principle, to evaluate whether the defects in DNA damage/repair and the consequent genetic heterogeneity that we observed in CD44+/CD24− cells were sufficient to increase both the phenotypic diversity and adaptability of a cancer cell population, we artificially induced cells into a CD44+/CD24− state and tested the cells' capabilities to adapt to different perturbations (i.e., exposure to different drugs).

We took advantage of the fact that when treated with TGF-$\beta$, A549 cells transit into a CD44+/CD24− state (i.e., differential surface expression of CD44 and CD24, increased expression of *VIM*, *SNAI1* and *IL-6* and decreased expression of both E-cadherin (*CDH1*) and the HDR genes described previously), but upon TGF-$\beta$ withdrawal, they return to their original epithelial state (*Figure 7A and B* and *Figure 7—figure supplement 1*). We found that transient exposure to TGF-$\beta$ results in an increased genetic diversity (*Figure 7C*). The reversibility of the CD44+/CD24− cell state was crucial to test our hypothesis as it enabled us to determine whether an increased drug resistance was due

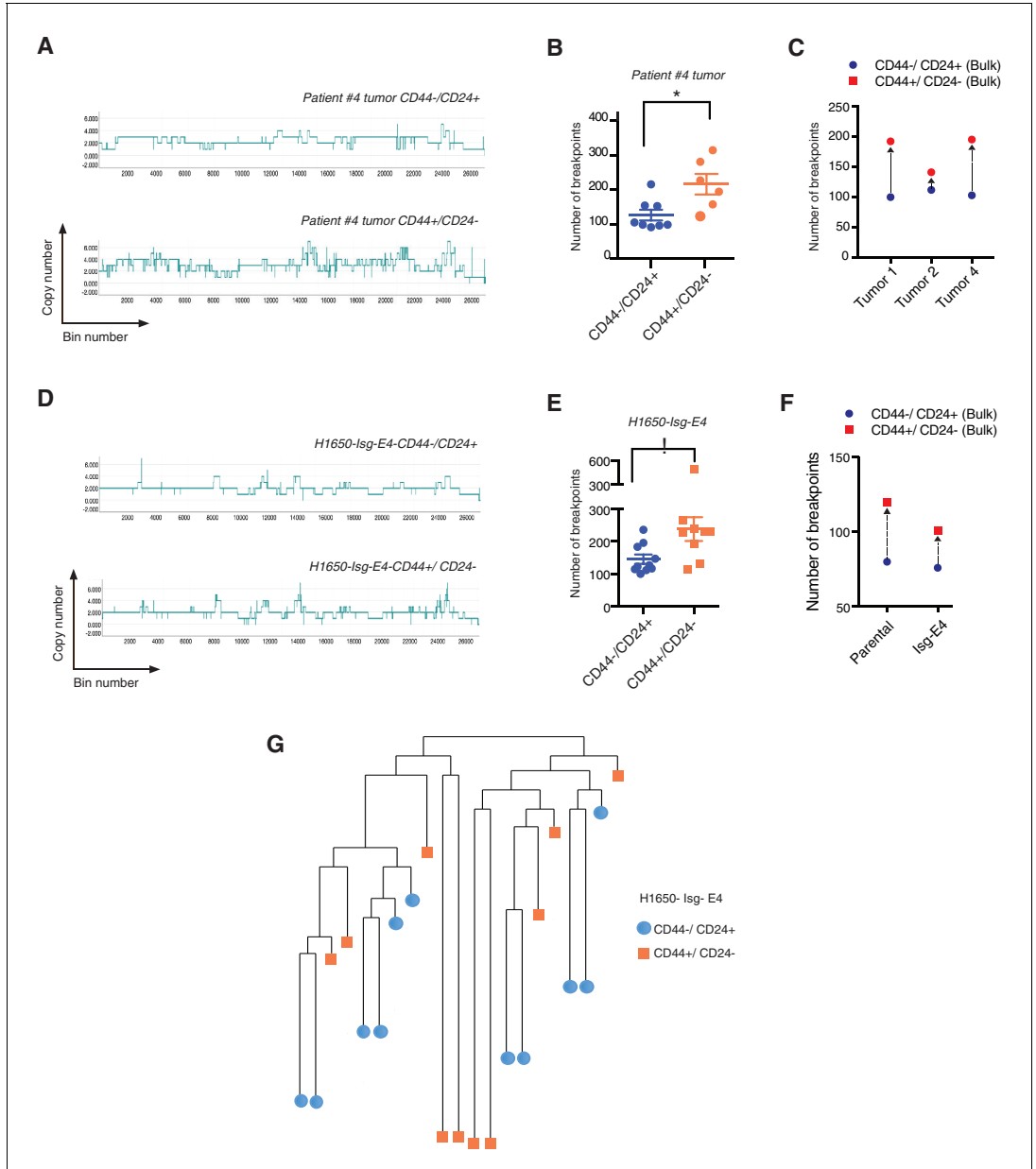

**Figure 5.** CD44+/CD24− cells have higher copy number alterations and increased genetic diversity. (A) The graph illustrates a representative copy number profile of CD44−/CD24+ cells and CD44+/CD24− cells sorted from an NSCLC patient (Patient #4). The x-axis corresponds to bins across the genome space from chr1 on the left to the sex chromosomes on the right. The y-axis corresponds to the copy number value at each bin. (B) The chart represents the number of DNA joint points in FACS-sorted CD44−/CD24+ cells (blue circle) and CD44+/CD24− cells (orange circle) from the indicated human NSCLC tumor. Each dot represents the analysis of a single cell. The breakpoint matrix (utilized to calculate DNA joint points), cluster dendogram and heat-map of normalized read counts (*Figure 5—figure supplement 1A*) were generated using Ginkgo, an open-source web platform for interactive analyses of CNA. (*Garvin et al., 2015*) (http://qb.cshl.edu/ginkgo). A variable bin size of 175 kb is used. p-value * < 0.05, unpaired t-test with Welch's correction. Error bars indicate standard deviation. (C) The chart represents number of DNA joint points in CD44−/CD24+ (blue circle) and CD44+/CD24− (red circle) cells FACS-sorted from the indicated primary human NSCLC. Each dot represents the analysis of a cell type collected and sequenced in bulk. (D) The graph illustrates a representative copy number profile of one CD44−/CD24+ and one CD44+/CD24− FACS-sorted cell from the H1650-derived isogenic cell line H1650-Isg-E4. (E) The chart represents the number of DNA joint points in FACS-sorted CD44−/CD24+ cells (blue circle) and CD44+/CD24− cells (orange squares) from the H1650-Isg-E4 cell line. Each dot represents the analysis of a single cell. The breakpoint matrix (utilized to calculate DNA joint points) is generated along with the cluster dendogram and heat-map of normalized read-counts (*Figure 5—figure supplement 1B*) using Ginkgo. A variable bin size of 175 kb is used. p-value *<0.05, unpaired t-test with Welch's correction. Error bars indicate standard deviation. (F) The chart depicts the number of DNA joint points in CD44−/CD24+ (blue circle) and CD44+/CD24− (red square) cells FACS-sorted from the H1650 (parental) and H1650 isogenic Isg-E4 cell lines. Each dot represents the analysis of a cell type collected and sequenced in bulk.

*Figure 5 continued on next page*

*Figure 5 continued*

(**G**) Cluster dendogram of normalized read-counts across segment breakpoints (using Euclidian distance and the ward-clustering method) of CD44−/CD24+ cells (blue circle) and CD44+/CD24− cells (orange squares) FACS-sorted from the H1650-Isg-E4 cell line. Each dot represents a single cell. The cluster dendogram is generated with Ginkgo.

The following figure supplement is available for figure 5:

**Figure supplement 1.** Heat-map of normalized read counts of FACS-sorted cells based on CD44 and CD24 surface markers from patient tumor and from the H1650-derived isogenic cell line H1650-Isg-E4.

to the effects of TGF-$\beta$ on genomic instability and clonal diversity, or to the acute and transient activation of drug-resistance signaling pathways.

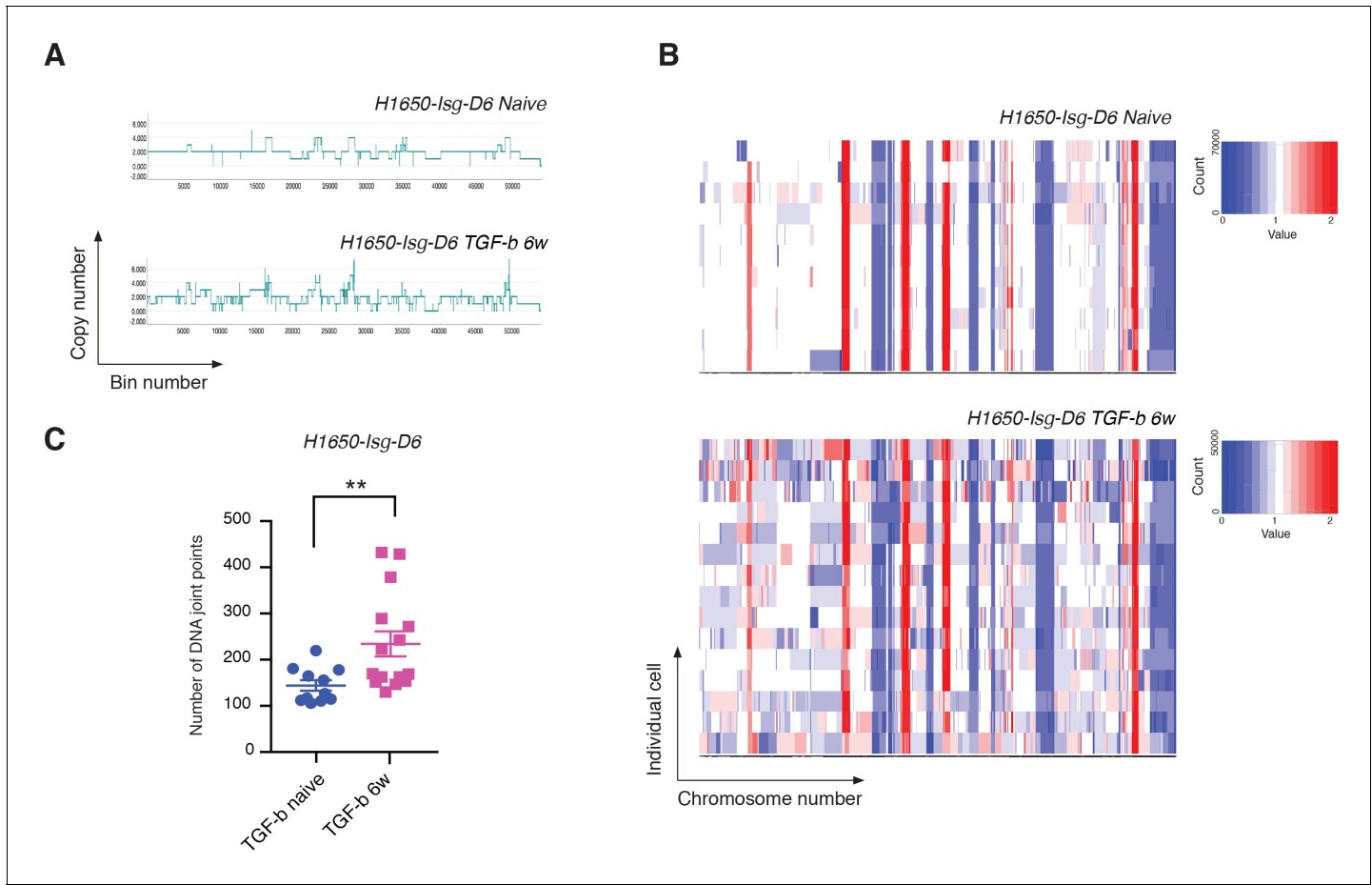

**Figure 6.** Exposure to TGF-$\beta$ is sufficient to increase CNA and genetic diversity of the cell population. (**A**) The graph illustrates a representative copy-number profile of one TGF-$\beta$-naïve cell and one TGF-$\beta$-treated (TGF-b 6w) cells from the H1650 isogenic/single cell-derived cell line Isg-D6. H1650-Isg-D6 was treated with vehicle (DMSO) or TGF-$\beta$ (1 ng/ml of each of TGF-$\beta$1 and -$\beta$2) for six weeks. Single cell CNA analysis was performed upon TGF-$\beta$ withdrawal. The x-axis corresponds to bins across the genome space from chr1 on the left to the sex chromosomes on the right. The y-axis corresponds to the copy number value at each bin. (**B**) Heat map of normalized read counts of TGF-$\beta$-naïve and TGF-$\beta$-treated (TGF-b 6w) cells from the H1650-Isg-D6 cell line, across segment breakpoints with a variable bin size of 50 kb (using Euclidian distance and the ward clustering method). Each horizontal line across the y-axis represents an individual cell, whereas the x-axis annotates the CNA across chromosomes from chr1 on the left to the sex chromosomes on the right. (**C**) The chart represents the number of DNA joint points in TGF-$\beta$-naïve (blue circle) and TGF-b 6w (purple squares) cells from the H1650-Isg-D6 cell line. Each dot represents the analysis of a single cell. The breakpoint matrix (utilized to calculate DNA joint points) is generated along with the cluster dendogram and heat-map of normalized read-counts (as shown in (**B**)) using Ginkgo with a variable bin size of 50 kb (http://qb.cshl.edu/ginkgo). p-value **=0.0064, unpaired t-test with Welch's correction.

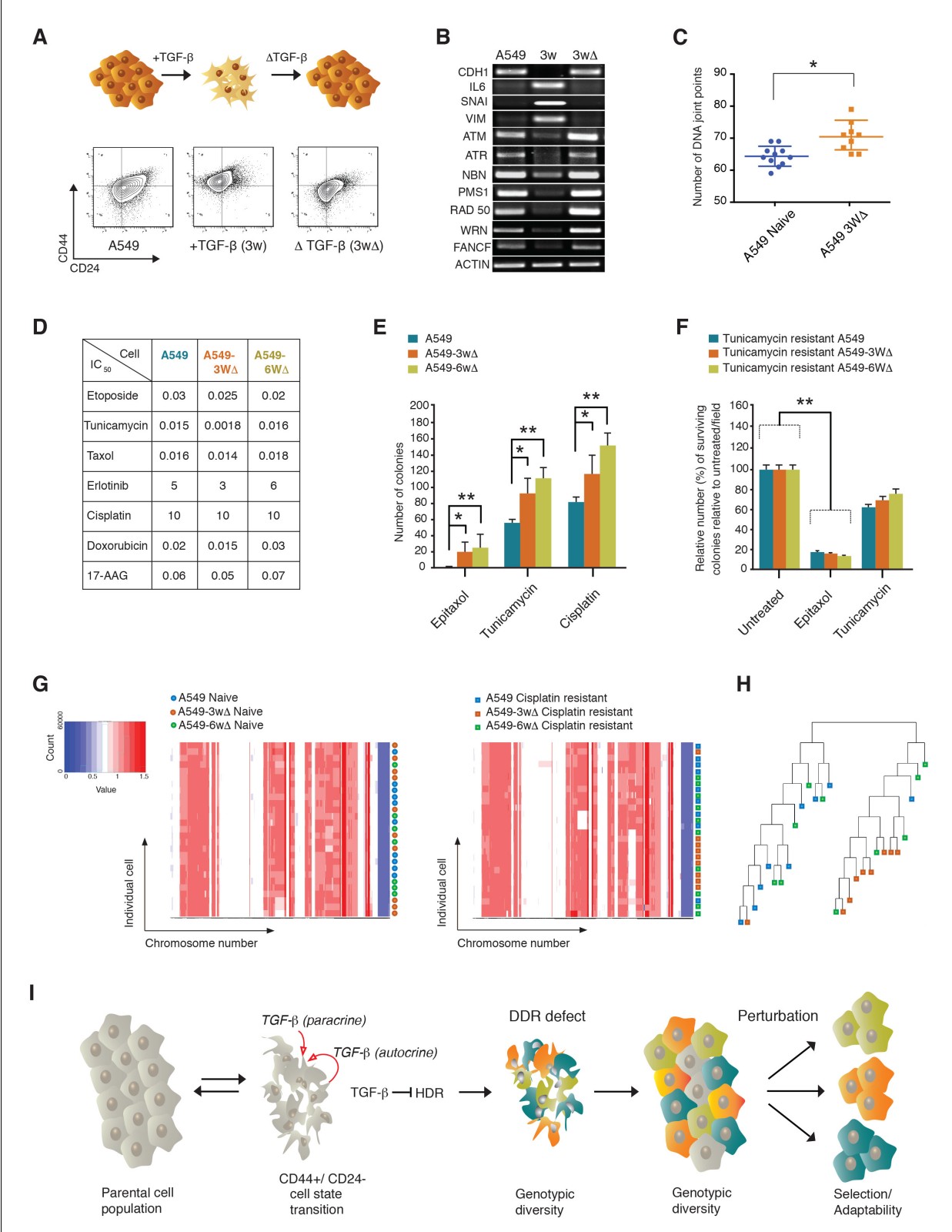

**Figure 7.** The TGF-β-induced CD44+/CD24− cell state increases the adaptability of cell populations. (**A** and **B**) A549 cells exposed to TGF-β acquire phenotypic and molecular changes characteristic of a CD44+/CD24− cell state. Upon TGF-β withdrawal, the cells return to their original cell state, as indicated by the FACS analysis in (**A**) and the RT-PCR analysis in (**B**). See *Figure 7—figure supplement 1* for quantification of FACS analysis. (**C**) The chart indicates the number of DNA joint points in TGF-β-naïve and TGF-β-treated A549 cells. Each dot represents the analysis of a single cell. The

*Figure 7 continued on next page*

*Figure 7 continued*

breakpoint matrix (utilized to calculate DNA joint points) is generated using Ginkgo (http://qb.cshl.edu/ginkgo). A variable bin size of 175 kb is used. p-value *<0.05, unpaired t-test. (D) The table depicts the $IC_{50}$ values of A549, A549-3W△ and A549-6W△ cells in the context of treatment with the indicated drugs. (E and F) TGF-$\beta$ treatment increased the adaptability of cells. Cells that were transiently exposed to TGF-$\beta$ for 3 or 6 weeks (A549-3W△, A549-6W△) were then treated with the indicated drugs upon TGF-$\beta$ withdrawl. (E) The number of colonies (mean ± SD) that have survived epitaxol (1.6 μM), tunicamycin (3.2 μM) and cisplatin (1 mM) treatment. Notably, the concentration of drugs used in this experiment corresponds to approximately >100X the $IC_{50}$. Two independent experiments, each with three replicates, were carried out and approximately 50 fields were counted for each sample. (p-value *<0.05, **<0.005, paired t-test.) (F) A549 tunicamycin-resistant clones were grown in regular/drug-free medium for a week and then retested for sensitivity to tunicamycin (3.2 μM) or epitaxol (1.6 μM). The plot represents mean ± SD number of colonies surviving 5 days after treatment compared with untreated cells, from two independent experiments each with three replicates (p-value **<0.005, paired t-test). (G) Heat map of normalized read counts across segment breakpoints (using Euclidian distance and the ward-clustering method) of the indicated cells. Each horizontal line across the y-axis represents an individual cell, whereas the x-axis annotates the CNA across chromosomes from chr1 on the left to the sex chromosomes on the right. A heat map of cisplatin-naïve cells is shown on the left and of cisplatin-resistant cells on the right. (H) Cluster dendogram of normalized read counts across segment breakpoints (using Euclidian distance and the ward-clustering method) of cisplatin-resistant-A549, cisplatin-resistant-A549-3W△ and cisplatin-resistant-A549-6w△. The cluster dendogram and heat-map of normalized read counts were generated using Ginkgo. (I) Schematic of proposed model. When cells transit into a CD44+/CD24− state, they acquire mesenchymal-like features and autocrine secretion of TGF-$\beta$ that leads to the downregulation of HDR genes. This process results in a hyper-mutable phenotype that spurs genetic diversity and intra-tumor clonal heterogeneity. Consequently, following a Darwinian model of cancer evolution, the transition of cancer cells into a CD44+/CD24− state or exposure to TGF-$\beta$ leads to an increased adaptability to any given perturbation.

The following figure supplement is available for figure 7:

**Figure supplement 1.** A549 cells undergo changes in the surface expression of CD44 and CD24 before, during and after exposure to TGF-$\beta$.

When we treated A549 TGF-$\beta$-naïve cells and A549 cells that were transiently exposed to TGF-$\beta$ for 3 weeks and 6 weeks (A549-3w△ and A549-6w△, respectively) with different concentrations of epitaxol, cisplatin, doxorubicin, erlotinib and tunicamycin, we found that they had similar half maximal inhibitory concentrations (IC50s) (*Figure 7D*) This was not the case when cells were treated with very high concentrations of tunicamycin, epitaxol and cisplatin. In this experiment, the number of colonies that could survive the drug treatment was in fact consistently and significantly higher for A549-3w△ and A549-6w△ cells (*Figure 7E*).

In principle, the increased adaptability that we observed in A549-3w△ and A549-6w△ cells upon exposure to high drug concentrations could be explained by the selection of a particular clonal population during TGF-$\beta$ treatment. If this were the case, cells that were resistant to one drug would be equally resistant to the other drug treatments. But when we tested the sensitivity of tunicamycin-resistant clones to tunicamycin and taxol, we observed that these cells retained resistance to tunicamycin but were still sensitive to taxol. This result indicated that any particular clonal population was not selected but rather presented the possibility that their resistance was due to diverse molecular mechanisms (*Figure 7F*). Whole genome sequencing of cisplatin-resistant clones further supported this likelihood (*Figure 7G and H*).

## Discussion

Individual malignant cells within a tumor can possess a wide variety of traits (e.g., affecting growth, metabolism, motility, morphology, stress responses, etc.) that could ultimately impact the progression and recurrence of cancer (*Greaves and Maley, 2012*; *Almendro et al., 2013*; *Burrell et al., 2013*; *Kreso and Dick, 2014*). The acquisition of these distinct features may result from genetic mutations and/or non-genetic determinants, such as the fluctuation among different cell states resulting from epigenetic/stochastic mechanisms or exposure to cues present in the micro-environment.

Here, we have shown that cells that reside in a CD44+/CD24− state are characterized by intrinsic defects in their abilities to repair DNA DSBs, leading to both augmented genetic instability and clonal diversity. Our results demonstrate, for the first time, that the oscillation between different cell states is not 'genetically' neutral. Instead, much like stress-induced mutagenesis, the oscillation between cell states can both promote the acquisition of new mutations and increase the total phenotypic and genomic diversity of a cancer population (*Meng et al., 2005*; *Ponder et al., 2005*;

*Chan et al., 2008*; *Shee et al., 2011*; *Gutierrez et al., 2013*; *Fitzgerald et al., 2017*). Thus, the CD44+/CD24− state not only can serve as a substrate for but also can spur tumor evolution by promoting continued acquisition of genetic diversity.

When we consider that cancer cells within a tumor can die, proliferate, enter dormancy and/or exhaust their long-term clonal growth, then therapy failure and reoccurrence should be attributed not only to the acquisition of new mutations, but also to the surviving cells' ability to propagate clonally. Consequently, the tumor cells contributing to recurrence must have regenerative potential and behave like cancer stem cells. In light of these considerations, the fact that CD44+/CD24− cells have self-renewal capabilities and have been referred to as 'cancer stem cells' is particularly captivating (*Al-Hajj et al., 2003*; *Mani et al., 2008*; *Korkaya et al., 2011*; *Brooks et al., 2015*). Another interesting feature of CD44+/CD24− cells is their general sturdiness and ability to survive exposure to lethal stimuli when compared to cells in other cell states (*Yao et al., 2010*). As noted above, CD44 +/CD24− cells express higher levels of genes that are known to protect cells from pro-apoptotic stimuli, including *BCL-2*, *BCL-XL* and *MCL-1* (*Keitel et al., 2014*). It is, therefore, tempting to propose that the transition of cancer cells into a CD44+/CD24− state could provide not only a small sub-population of tumor cells that can withstand and survive an initial destructive drug attack, but also the means to accumulate new genetic mutations that will further increase their fitness, ultimately allowing their growth and expansion even in the presence of a drug.

Our data indicated that TGF-$\beta$-mediated signaling is both necessary and sufficient to reduce the expression of HDR genes and to increase both CNA and the genetic heterogeneity of cancer cell populations (*Figures 6* and *7*). TGF-$\beta$ is not only produced by CD44+/CD24− cells but it is among the many factors that mediate the communication between the cancer cells and their surrounding stroma (*Massagué, 2008*). Hence, our findings are particularly exciting because, in principle, they could provide additional support for the role of the tumor microenvironment in shaping tumor fitness. This could be especially relevant in the case of chronic inflammatory conditions and current cancer treatments, as both have been shown to increase TGF-$\beta$ levels in tissues. Consequently, we can view chemotherapy regimens as double-edged swords. While they can kill cancer cells by inducing inflammation and TGF-$\beta$, they can also spur cancer cell evolution and paradoxically contribute to drug resistance. In other words, as noted by Dr Huang in one of his current review articles, our observations may echo Nietzsche's classic statement: 'What does not kill me, makes me stronger' (*Pisco and Huang, 2015*). This could be particular relevant when designing cancer treatments because the progression-stimulating effects of a therapy could shape the behavior of the tumor (*Pisco and Huang, 2015*).

Whereas exposure to TGF-$\beta$ in malignant cells often elicits pro-tumorigenic activities, in non-transformed cells, it can instead induce cell cycle arrest, senescence and apoptosis through a p53-, p21- and p16-dependent mechanism (*Massagué, 2008*). As persistent DSB could elicit these same effects, the downregulation of DNA-repair genes and the consequent induction of DNA damage by TGF-$\beta$ in non-transformed cells could be part of the tumor-suppressor activities of TGF-$\beta$. As often is the case in cancer, a mechanism that has been implemented during evolution to protect tissue homeostasis could become hijacked by cancer cells to promote their survival. In this regard, the observation that the p53 tumor suppressor network is disabled in cancer cells is particularly interesting.

TGF-$\beta$ plays a role not only in the context of tumorigenesis but also during development and in adult organism by regulating tissue-injury responses, adaptive and innate immune responses, tissue homeostasis and so on (*Massagué and Gomis, 2006*). The possibility that TGF-$\beta$ could impair DNA-damage responses not only in the context of cancer cells but also in normal somatic cells is intriguing. This, in fact, could provide a means to activate a tumor suppressor network that is based on DNA-damage responses. Consequently, TGF-$\beta$=mediated impaired HDR could be part of an intrinsic cell mechanism that balances cell proliferation in tissues. On the other hand, it could also be possible that when DNA-damage-repair mechanisms are dimmed, the genetic diversity of cells in tissues is increased. Herein, inhibition of HDR by TGF-$\beta$ could provide a means to increase the heterogeneity of cells within tissues during development. This phenomenon could be important in the context of organs or tissues in which diversity could increase functionality as, for example, in the case of immune cells or neurons (*Rehen et al., 2001*, *2005*). Interestingly in this regard, recent data have indicated that certain somatic cells are not genetically uniform but are characterized by different ploidy (*Duncan et al., 2010*, *2012*).

HDR-mediated repair is also utilized by viruses and transposons (*Yant and Kay, 2003*). Hence, the downregulation of component of this DNA-repair pathway by TGF-β could be part of innate mechanisms that limit the reactivation of transposable elements and viral integration.

Notably, some reports have illuminated the possible role of TGF-β in regulating the expression and/or activities of DNA-repair genes. Similar to our observations, (*Kanamoto et al., 2002*) showed TGF-β-dependent downregulation of Rad51 and decreased DNA -epair efficiency in Mv1Lu lung epithelial cells treated with TGF-β. In addition, *Liu et al. (2014)* reported that TGF-β regulates the expression and/or activity of *BRCA1*, *ATM* and *MSH2*. Conversely, *Glick et al. (1996)* reported that mouse keratinocytes deficient in TGF-β show a significant increase in gene amplification in response to the drug PALA. *Kirshner et al. (2006)* also showed that TGF-β signaling inhibition attenuates the function of ATM under stress (ROS stimulation), thus impeding the response to DNA damage. *Wiegman et al. (2007)* provided evidence that TGF-β can stimulate ATM and p53 phosphorylation in primary irradiated cells in a Smad-independent pathway. Our multi-faceted approaches and data gathering greatly expand upon this existing body of evidence and strongly indicate that TGF-β plays a broader role in decreasing the expression of DNA-repair genes, contributing to CNA accumulation and the clonal diversity of cancer cell populations. As a possible explanation for the differences observed across the studies mentioned here, we noted that those indicating that TGF-β promotes DNA-damage-repair responses were conducted under stress conditions (i.e., PALA treatment, irradiation or $H_2O_2$ treatment) and in response to reactive oxygen species (ROS) and did not distinguish between the effect of TGF-β on these experimental conditions (i.e., on ROS scavengers) and a direct effect on DNA repair. Our findings are also in accordance with a recent review on the implications of stress-induced mutagenesis (*Fitzgerald et al., 2017*).

In summary, our data strongly indicate that the transition into a CD44+/CD24− cell state can promote intra-tumor genetic heterogeneity, spur tumor evolution and increase tumor fitness. These findings have important prognostic and therapeutic implications and speak strongly to the need to target CD44+/CD24− populations in tumors. In this regard, the observation that the decreased expression of HDR genes in CD44+/CD24− cells comes at the expense of increased dependency on other DNA-repair components is particularly notable (*Figures 1* and *2*). As some of these genes are potentially druggable, our studies could apprise the development of novel therapeutic options to ameliorate the outcome of cancer based on targeting the processes of cancer evolution, instead of targeting the cancer evolution products.

# Materials and methods

## Cell culture

A549 (RRID: CVCL_0023), H1650 (RRID: CVCL_1483), HCC4006 (RRID: CVCL_1269), H23 (RRID: CVCL_1547), BT-474 (RRID: CVCL_0179), MDA-MB-231 (RRID: CVCL_0062), MDA-MB-435S (RRID: CVCL_0622) and NCI-H23 (RRID: CVCL_1547) were obtained from the American Type Culture Collection repository (ATCC, Manassas, Virginia). The MCF-7 (RRID: CVCL_0031) cell line was obtained from the Cold Spring Harbor Laboratory Tissue Culture Facility. The PC9 (RRID: CVCL_B260) cell line was a gift from Dr Jeffrey A. Engelman (MGH, Charlestown, MA). The U2OS-DR-GFP cell line was created by Dr Maria Jasin and was a gift from Dr Agata Smogorzewska (Rockefeller University, NY). The NMuMG (RRID: CVCL_0075) cell line was a gift from Linda Van Aelst (CSHL, NY). The A549, H1650, HCC4006, U2OS, H23 and PC9 cell lines were cultured in RPMI medium supplemented with 10% FBS, glutamine, penicillin and streptomycin. MCF7, BT-474, MDA-MB-231 and MDA-MB-435s were cultured in DMEM containing 10% FBS, penicillin, streptomycin and sodium pyruvate. NMuMG was cultured in complete DMEM medium supplemented with 10 µg/ml bovine insulin. These cell lines were monitored for mycoplasma contamination by using the Lonza mycoalert mycoplasma detection kit on a regular basis. All of the cell lines tested negative for *Mycoplasma* contamination. None of the cell lines used in our studies was mentioned in the list of commonly misidentified cell lines maintained by the International Cell Line Authentication Committee.

## Quantitative RT-PCR (qRT-PCR)

RNA was isolated from cells using the TRIzol-chloroform method. Cells were lysed with TRIzol reagent (Invitrogen, Carlsbad, CA). After the chloroform extraction, the aqueous phase containing

RNA was mixed with a 0.7 vol of isopropanol and centrifuged at 13,000 rpm for 15 min to precipitate the RNA. Following a wash with 70% ethanol, the RNA pellet was washed with 70% ethanol, air dried, dissolved in distilled water and subjected to DNAseI treatment to remove any contaminating genomic DNA. The quality and concentration of DNA and RNA were assessed using a Nanodrop ND-1000 spectrophotometer (Nanodrop). cDNA was prepared using the ImProm-II cDNA synthesis kit (Promega, Madison, WI). qPCR was carried out using the Power SYBR Green PCR Master MIX (Applied Biosystems, Carlsbad, CA) with 7900HT Fast Real-Time PCR system (Applied Biosystems). The primers used to measure mRNA expression at the cDNA levels by RT-qPCR are listed in *Supplementary file 3*.

## Transfection and cell survival assay

Cells were seeded in six-well plates and transfected with siRNA or plasmid or both with Lipofectamine reagent 2000 (Invitrogen) for at least 6 hr in antibiotic-free culture media mixed with Opti-MEM glutaMax media (Gibco). After transfection, the medium was changed to DMEM or RPMI (as described above) supplemented with FBS and antibiotics. At 72 hr post-transfection, the cells were harvested for RNA or protein analysis. To check for apoptosis, the media from the culture vessels were also collected and the dead cells were collected by centrifugation. For survival assays, the cells were trypsinized and counted at least 4 hr after changing the transfection medium. Depending on the cell line, between 2,000 and 3,000 cells were plated in a 96-well plate and cultured for 96–120 hr. Cells were then washed with PBS once and fixed in 3.7% formaldehyde (Thermo Fisher Scientific, Waltham, MA) at room temperature for 12 min. Next, cells were stained with Syto-60 Red fluorescent nuclein acid stain (Thermo Fisher Scientific) and scanned with ODYSSEY infrared imager (LI-COR, Lincoln, NE) at 700 nm.

## siRNA-mediated gene expression knockdown

One hundred picomoles (pmol) of Stealth RNAi (Invitrogen) siRNA against human-*IL-6*, -*BRCA1*, -*ORC5L*, -*RFC3*, -*RPA2*, -*NEK9*, -*POLS*, -*ERCCC8*, -*SMAD2*, -*SMAD3*, -*SMAD4*, -*RELA*, -*RFC1*, -*ATAD5*, -*CTF18* and -*RAD17* or 40 pmol of Silencer Select (Ambion) siRNA against human-*BRCA2* and -*RAD50* or 20 pmol of ON-TARGET plus SMART pool (Dharmacon, Lafayette, Colorado) against human *BLM* were used in six-well plates. The siRNA sequences (sense strands) are listed below:

    ORC5HSS181666.1 – CGU UUG UCU UAU AUU UCC CUG AUU A
    ORC5HSS181666.2 – UAA UCA GGG AAA UAU AAG ACA AAC G
    RFC3HSS184273.1 – AAG GCU GUA UGA GCU UCU AAC UCA U
    RFC3HSS184273.2 – AUG GAU UAG AAG CUC AUA CAG CCU U
    RPA2HSS184377.1 – CAG AAU UGG GAA UGU UGA GAU UUC A
    RPA2HSS184377.2– UGA AAU CUC AAC AUU CCC AAU UCU G
    ERCC8HSS174455.1 – AGC AGU UUC CUG GUC UCC ACG UUA U
    ERCC8HSS174455.2 – AUA ACG UGG AGA CCA GGA AAC UGC U
    PAPD7HSS117093 (POLS).1 – CCU UGG AAU GCU UCU UGU AGA AUU U
    PAPD7HSS117093 (POLS).2 – AAA UUC UAC AAG AAG CAU UCC AAG G
    IL6HSS105337.1 – GAG AAA GGA GAC AUG UAA CAA GAG U
    IL6HSS105337.2 – ACU CUU GUU ACA UGU CUC CUU UCU C
    BRCA1HSS101089.1 – GGG CUA UCC UCU CAG AGU GAC AUU U
    BRCA1HSS101089.2 – AAA UGU CAC UCU GAG AGG AUA GCC C
    BRCA2.1 –GGA UUA UAC AUA UUU CGC Att
    BRCA2.2 –CAG UUG AAA UUA AAC GGA Att
    RAD50.1 –GGU AGA CUG UCA UCG UGA Att
    RAD50.2 –GGA AUA GAC UUA GAU CGA Att

## TGF-β treatment

$1.5 \times 10^6$ cells were plated in a complete media in 10 $cm^2$ tissue culture dish on day 1. On day 2, the medium was changed to –Serum RPMI (or DMEM) and the cells were starved overnight. On the morning of day 3 (40 hr post seeding), cells were treated with rhTGF$\beta$1 and rhTGF$\beta$2 (R&D systems,

Minneapolis, MN) 1 ng/ml each in complete media, for 9 hr. Following treatment, the cells were harvested for RNA preparation and qRT-PCR, for immunoblotting or for cell cycle analysis.

## Drug treatment

For treatment with LY2157299 (20 µM) and LY364947 (1 µM) (TGFBR1 kinase inhibitor, Selleckchem, Houston, TX), 300,000 H1650-M3 cells were plated in a 6 cm² plate. Inhibitor was added the next day and the mixture was incubated for 3–5 days for LY2157299 and 2–3 days for LY364947. The cells were lysed with TRIzol and processed for RNA preparation. To determine IC50 values for various drugs (17-AAG (this drug is not mentioned in main text), cisplatin, doxorubicin, etoposide, erlotinib, epitaxol and tunicamycin), the cells were plated in 96-well plates at 2,000 cells/well. The next day, individual drugs were added to the wells at the indicated concentrations and the mixture was incubated for 5 days. The plates were then washed once with PBS, fixed with 3.7% formaldehyde and stained with crystal violet. Each stained well was destained in 50–100 µl of 10% acetic acid and the absorbance was read in a spectrophotometer at 590 nm.

## Flow cytometry

The cells were trypsinized using TrypLE, incubated at 37°C and re-suspended in PBS containing 1% FBS. Depending on the sample, between 800,000–1,000,000 cells/sample were added to a MACS buffer (PBS with 0.5% BSA and 2 mM EDTA). The cells were washed in the MACS buffer twice and re-suspended in 100 µl staining solution (100 µl MACS buffer + antibodies). Staining was performed for 45 min and the cells were then washed twice with MACS buffer and re-suspended in 400 µl MACS buffer before passing through a 5 ml polystyrene round-bottom tube with cell-strainer cap (BD-Falcon, Corning, New York). Samples were analyzed with the BD LSR II Cell Analyzer. For the cell-sorting procedure, the samples were prepared as above and sorted with the BD FACS Aria (SORP) Cell Sorter. Samples were collected and either cultured on an eight-well chamber slide system (LAB-TEK, Thermo Fisher Scientific) for 2 days before fixing and staining or subjected to RNA preparation. For the cell-cycle analysis, trypsinized cells were washed in MACS buffer and then fixed in ice-cold 70% ethanol by dropwise addition while vortexing, followed by washing in MACS buffer and staining $1 \times 10^6$ cells in PI solution (PBS with 50 µg/ml propidium Iodide, 0.1 mg/ml RNase A, 0.05% TritonX-100) for 45 min at 37°C, passed through a 5 ml polystyrene round-bottom tube with cell-strainer cap (B-Falcon) and run on the BD LSR II Cell Analyzer. Cell cycle arrest was assessed through analysis of the proportion of cells in the G1, S and G2/M fraction of the cell cycle.

## Immunoblotting

Cells were washed with PBS before collection and lysed directly in RIPA buffer containing 0.2% SDS for 30 min on ice. Proteins were separated by 6–12% SDS/PAGE, transferred to nitrocellulose membranes (Bio-Rad, Hercules, CA) and blotted with antibodies as indicated. To extract BRCA2, RAD50, BLM and WRN, cells were lysed in ice-cold NETN-450 buffer (450 mM NaCl, 1 mM EDTA, 20 mM Tris pH8 and 0.5% Igepal CA-630) for 15 min on ice followed by extraction in NETN-0 (1 mM EDTA, 20 mM Tris pH8 and 0.5% Igepal CA-630) for 15 min on ice.

## Immunofluorescence

CD44+/CD24− cells and CD44−/CD24+ cells from H1650 were FACS-sorted and cultured for 2 days in an eight-well chamber slide system (LAB-TEK, Thermo Fisher Scientific). H1650, A549 and MCF7 cells were grown on glass coverslips in a 24-well Petri dish and treated with TGF-$\beta$ or vehicle (DMSO) for 3 and 4 days or with 20 µg/ml doxorubicin overnight. Cells were fixed with 3.7% formaldehyde and permeabilized in 0.1% Triton X-100 in PBS for 10 min. Fixed cells were washed three times in PBS and blocked with 1% BSA in PBS for 1 hr. After washing three times with PBS, the cells were incubated with the primary antibody for overnight at 4°C. Immune complexes were then stained with indicated secondary antibodies (Invitrogen). DAPI was used for nuclear staining. The stained cells were mounted with a Vectashield mounting medium (Vector Laboratories, Burlingame, CA) and analyzed using a confocal microscope.

## Comet assay

Cells embedded in agarose on a microscope slide were lysed with detergent and high salt to form nucleoids containing supercoiled loops of DNA linked to the nuclear matrix. Electrophoresis at high pH results in structures resembling comets that can be observed by fluorescence microscopy. The intensity of the comet tail relative to the head reflects the number of DNA strand breaks.

## DR-GFP assay to measure HR efficiency

DR-GFP consists of two mutated GFP genes: (i) the Sce-GFP that is disrupted by an 18-bp recognition site for the I-SceI endonuclease and (ii) the iGFP that is truncated at the 5′ and 3′ ends. Upon transfection with I-SceI, the SceGFP is cleaved and, after an HDR event has occurred, a functional GFP+ gene (detectable by flow cytometry) is generated by gene conversion with iGFP. To perform a DR-GFP assay in U2OS cells, U2OS-DR-GFP cells were plated in six wells and treated with vehicle (DMSO) or TGF-$\beta$ (1 ng/ml each of TGF-$\beta$1 and TGF-$\beta$2) for a day. Then, cells were transfected with 4 ug of pCBASce-I (Jasin Lab) or control plasmid (pCAG:mRFP1, data not shown) for 6 hr, transfection media were removed and cells were grown in media with vehicle (DMSO) or TGF-$\beta$ (1 ng/ml each of TGF-$\beta$1 and TGF-$\beta$2) for 3 days. Cells were then harvested and subjected to FACS analysis. To perform a DR-GFP assay in H1650 cells, the cells were plated in triplicate in six wells and treated with vehicle (DMSO) or TGF-$\beta$ (1 ng/ml each of TGF-$\beta$1 and TGF-$\beta$2) for a day. Then, cells were transfected with 0.5 μg of pDR-GFP (Addgene) or 2 μg of pCBASce-I (Jasin Lab) or both the plasmids for 6 hr; subsequently, transfection media were removed and cells were grown in media with vehicle (DMSO) or TGF-$\beta$ (1 ng/ml each of TGF-$\beta$1 and TGF-$\beta$2) for 3 days. Cells were then harvested and subjected to FACS analysis. 20,000 cells were analysed for each sample replicate.

## Generation of isogenic clones

H1650 cells were serially diluted in 96 wells such that one well contains one cell. They were then grown for 2 months before the experiments.

## Single nuclei sorting (SNS)

To perform SNS, cells from culture were stained with antibodies against CD44 and CD24 or nuclei were isolated from cells in culture and stained with 49,6-diamidino-2-phenylindole (DAPI). We use FACS to gate a desired population of cells (for antibody-stained cells , we gated CD44−/CD24+ and CD44+/ CD24−) or nuclei (by total DNA content) and deposited desired cells/nuclei singly into 96-well plates. For sorting cells from tumor, we stained a single-cell suspension derived from a tumor with CD45, CD31, EpCAM, CD44 and CD24 antibodies. We gated the CD45−/ CD31−/ EpCAM $^{mid/high}$ population and then gated the desired CD44−/CD24+ and CD44+/ CD24− from the EpCAM $^{mid/high}$ population. After WGA using Sigma GenomePlex, sonication was performed to create free DNA ends without WGA adapters and then we constructed libraries for 76-bp, single-end sequencing using one lane of an Illumina GA2 flow cell per nucleus. For each nucleus, we typically achieve at least 9 million (mean 59.042 million, s.e.m. 60,328, n = 5,200) uniquely mapping reads using the Bowtie alignment software. These sequences cover approximately 6% (mean 5.95%, s.e.m. 0.229, n = 5,200) of the genome and are used to count sequence reads in 50,000 variable bins. The bin counts are segmented using a KS statistic and then used to calculate integer copy number profiles. Neighbor-joining trees are constructed both from the integer profiles and from the chromosome breakpoint patterns of each cell to infer evolution.

## Single-cell sequencing data analysis using Ginkgo

The copy-number profiles, dendrograms and heatmaps of single cells were generated using Ginkgo, a visual analytics tool for the analysis of single-cell copy-number variations. Ginkgo automatically corrects for known biases in single-cell data and generates copy number profiles for each cell. To generate copy number profiles, we used Ginkgo with the the following parameters: variable length 500 kb bins simulated using 101 bp reads mapped with bowtie, cell segmented independently with normalized read counts and all bad bins masked. In order to cluster cells to generate dendrograms and heat maps, we using ward linkage and euclidian distance metrics. Quantification of DNA joint points (referred to as 'breakpoints' in the Ginkgo website) was done by extracting the breakpoint counts for each cell from the 'Breakpoints' file under Ginkgo's 'Download processed data tab'.

## Drug adaptability assay

Twenty-five thousand cells were plated in each well of a six-well plate and treated with either 3.2 μM tunicamycin (Sigma-Aldrich, St. Louis, MO), 1.6 μM epitaxol (Santa Cruz Biotechnology, Santa Cruz, CA) or 3.2 μM etoposide (Sigma-Aldrich) for 3 days. In another assay, 12,000 cells were plated in a six-well plate; after the cells formed small colonies with ~10 cells/ colony, they were treated with 3.2 μM tunicamycin for 3 days followed by either tunicamycin or epitaxol (1.6 uM) treatment for 3 days.

## Patient study details

The collection of human lung tissue samples and blood for this study was covered by Huntington Hospital/Northwell Health IRB #14–496 (PI V Singh; approval date 11/14/14). The samples were acquired from patients already undergoing thoracic procedures (e.g. surgical tumor resection or biopsy) at Huntington Hospital. All study participants provided informed consent for the use of their lung tissue and blood for research purposes. Participants were informed of study aims, the potential risks and benefits of participation, and that any discoveries facilitated by the analysis of their tissues might be published. The participants were informed that their names would not be associated their samples in any publication or presentation of research findings

## Antibodies used

The following antibodies were used for the flow cytometry analyses:

- PE/Cy7 anti-mouse/human CD44 antibody (BioLegend, San Diego, CA); cat. # 103029 (RRID: AB_830786)
- eFluor 450 anti-human CD24 antibody (eBioscience, San Diego, CA); cat # 48-0247-42 (RRID: AB_11218707)
- BV421 anti-human CD24 antibody (BD Biosciences, San Jose, CA); cat. # 562789
- PE-CF594 anti-human CD45 antibody (BD Biosciences); cat. # 562279
- APC anti-human CD31 antibody (eBioscience); cat # 17-0319-42 (RRID: AB_10852842)
- Alexa Fluor 488 anti-human CD326 (Ep-CAM) antibody (BioLegend); cat # 324210 (RRID: AB_756084).

The following antibodies were used for immunofluorescence:

- Anti-gamma H2A.X (phospho S139) antibody (Abcam, Cambridge, MA); cat. # ab11174 (RRID: AB_297813)
- Anti-53BP1 antibody (Abcam, Cambridge, MA); cat. # ab36823 (RRID: AB_722497)
- Alexa Fluor 488 donkey-anti-rabbit IgG (H+L) (Invitrogen); cat # A21206
- Alexa Fluor 568 donkey-anti-mouse IgG (H+L) (Invitrogen); cat # 10037

The following antibodies were used for immunoblot analysis:

- Cleaved caspase-3 (Asp 175) antibody (Cell Signaling Technology, Danvers, MA); cat. # 9661 (RRID: AB_2341188)
- Anti-alpha-tubulin antibody (Millipore, Billerica, MA); cat. # MABT205 (RRID: AB_11204167)
- Anti-BLM antibody (Atlas Antibodies, Bromma, Sweden); cat. # HPA005689 (RRID: AB_1845372)
- Anti-BRCA2 antibody (Atlas Antibodies); cat. # HPA026815 (RRID: AB_10602692)
- Anti-Rad50 antibody (Millipore); cat. # 05–525 (RRID: AB_309782)
- Anti-Ras-GAP antibody (BD Biosciences); cat. # 610040 (RRID: AB_397455)
- Anti-RDM1 antibody (Sigma-Aldrich, St. Louis, MO); cat. # HPA024794 (RRID: AB_1856039)
- Anti-WRN antibody (Cell Signaling Technology, Danvers, MA); cat. # 4666 (RRID: AB_10692114)

## Acknowledgement

This study was supported by the NCI P01 CA129243-06 target for therapy for carcinomas in the lung, the Elisabeth R. Woods Foundation and Swim Across America. We would like to acknowledge Ms Madison Miller and Drs Serif Senturk, Matthew Camiolo, Trine Lindsted and Robert Aboukhalil for technical assistance. We would also like to acknowledge the bioinformatics, next generation sequencing, microscopy, flow cytometry and core facilities at Cold Spring Harbor Laboratory.

## Additional information

### Funding

| Funder | Grant reference number | Author |
|---|---|---|
| National Cancer Institute | NCI P01 CA129243-06 | Raffaella Sordella |

The funders had no role in study design, data collection and interpretation, or the decision to submit the work for publication.

### Author contributions

DP, Data curation, Formal analysis, Validation, Investigation, Methodology, Writing—review and editing; AP, Data curation, Formal analysis, Validation, Investigation; NHS, Data curation, Formal analysis, Methodology, Writing—review and editing; ZY, Data curation, Formal analysis, Investigation, Methodology; NA, TG, JK, Resources, Software, Formal analysis, Methodology; HC, Resources, Data curation, Formal analysis, Methodology; KC, FR, VAS, Resources, Investigation, Methodology; LE, Resources, Supervision; GCS, Resources, Supervision, Validation; MCS, Conceptualization, Resources, Software, Supervision, Methodology; JH, Conceptualization, Resources, Supervision, Methodology; GJH, Conceptualization, Resources, Supervision, Investigation, Visualization, Methodology; RS, Conceptualization, Formal analysis, Supervision, Funding acquisition, Investigation, Writing—original draft, Writing—review and editing

### Author ORCIDs

Raffaella Sordella, http://orcid.org/0000-0001-9745-1227

### Ethics

Human subjects: Informed consent was received from all patients who participated in the study 14-496 (PI V Singh).

## Additional files

### Supplementary files

• Supplementary file 1. Mutation analysis of the following cells lines (used in our study) from COSMIC (Catalogue Of Somatic Mutations In Cancer). http://cancer.sanger.ac.uk/cosmic

• Supplementary file 2. Relative abundance of DNA-damage/-repair pathway components from gene expression profiles of H1650 and H1650-M3 cells.

• Supplementary file 3. Sequences of primers used in quantitative-RT-PCR in this study.

### Major datasets

The following previously published dataset was used:

| Author(s) | Year | Dataset title | Dataset URL | Database, license, and accessibility information |
|---|---|---|---|---|
| Cancer Genome Atlas Research Network, Weinstein JN, Collisson EA, Mills GB, Shaw KR, Ozenberger BA, Ellrott K, Shmulevich I, Sander C, Stuart JM | 2013 | The Cancer Genome Atlas Pan-Cancer analysis project | http://www.gitools.org/datasetfiles/tcga/TCGA-BRCA-gitoolsweb.heat-map.zip | Publicly available at Gitools (http://www.gitools.org) |

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
