## [Decision Letter]

Thank you for submitting your article "TGF-β reduces DNA ds-break repair mechanisms to heighten genetic diversity and adaptability of CD44+/CD24- cells" for consideration by *eLife*. Your article has been favorably evaluated by Kevin Struhl (Senior Editor) and three reviewers, one of whom is a member of our Board of Reviewing Editors. The reviewers have opted to remain anonymous.

The reviewers have discussed the reviews with one another and the Reviewing Editor has drafted this decision to help you prepare a revised submission.

As you will see from the reviewers' reports, there are differing opinions to what extent this manuscript presents substantial novelty, in how far the previous literature has been properly credited, and in how far this is a logical extension of previous research by your lab and other labs. In fact, this manuscript is presented as if all findings are new, and only in the second to last paragraph are previous observations mentioned "in passing". While we appreciate the logical flow of the experiments and how you came to the important conclusions based on the data provided, the manuscript would come over much better if you present right upfront what has already been learnt from previous publications, so that you and the reader can then better appreciate the novelty. This relates to previous relevant observations linking e.g. TGF-b signaling to increased chemoresistance, or TGF-b signaling to impaired DNA repair, or CSCs to DNA repair, etc. We do, however, fully appreciate the overall novelty and importance.

There were also comments that need to be experimentally addressed. Specifically:

It feels somewhat uncomfortable using only LY2157299 as TGF-b inhibitor, since it does have other signaling kinase targets besides the type I TGF-b receptor. Considering the central role of this aspect in the manuscript, further validation using another (more specific) TGF-b signaling inhibitor or neutralizing TGF-b antibodies would help. Even though LY2157299 is clinically under evaluation as galunisertib, further substantiation of specific TGF-b signaling would provide confidence.

The DR-GFP reporter system is used in U2OS cells (Figure 4 and supplementary data). These cells are not used in other experiments with major findings in this manuscript, are from mesenchymal origin, whereas the others are carcinomas, and have impaired TGF-b signaling. It would provide confidence to have this assay done in H1650 cells.

Additional comments to be addressed are:

The title and/or Abstract should provide a clear indication of the biological system under investigation (i.e., species name or broader taxonomic group, if appropriate). Please revise your title and/or Abstract with this advice in mind.

In all figures, please explain the error bars – what do they represent and if variance, how many experiments. Are the ones on the data from two experiments a range?

Throughout the paper where CNAs are discussed, please do not refer to "DNA breaks" or "break points". What is being quantifying is novel DNA junctions or join points. "Breaks" implies a mechanism of genome rearrangement that the authors do not know has occurred-one using DNA breakage. Models like microhomology-mediated replicative template switching and MMBIR are dominating modern understanding of CNA formation, and these are not break and join models. I suggest adopting either "join points" or new "DNA junctions", defining it for readers, and using throughout in all.

---

## [Author Response]

*As you will see from the reviewers' reports, there are differing opinions to what extent this manuscript presents substantial novelty, in how far the previous literature has been properly credited, and in how far this is a logical extension of previous research by your lab and other labs. In fact, this manuscript is presented as if all findings are new, and only in the second to last paragraph are previous observations mentioned "in passing". While we appreciate the logical flow of the experiments and how you came to the important conclusions based on the data provided, the manuscript would come over much better if you present right upfront what has already been learnt from previous publications, so that you and the reader can then better appreciate the novelty. This relates to previous relevant observations linking e.g. TGF-b signaling to increased chemoresistance, or TGF-b signaling to impaired DNA repair, or CSCs to DNA repair, etc. We do, however, fully appreciate the overall novelty and importance.*

In addressing this point, we have now substantially rewritten the Introduction and provided a better description of the current literature and background information.

*There were also comments that need to be experimentally addressed. Specifically:*

It feels somewhat uncomfortable using only LY2157299 as TGF-b inhibitor, since it does have other signaling kinase targets besides the type I TGF-b receptor. Considering the central role of this aspect in the manuscript, further validation using another (more specific) TGF-b signaling inhibitor or neutralizing TGF-b antibodies would help. Even though LY2157299 is clinically under evaluation as galunisertib, further substantiation of specific TGF-b signaling would provide confidence.

As suggested by the reviewers, we have now validated our finding with an additional TGF-β receptor1 inhibitor the LY364947 (Selleckchem). As shown in Figure 3—figure supplement 4, treatment of the CD44+/CD24- H1650M3 cells with LY364947 inhibitor for 72 hours was sufficient to reduce the expression of *SERPINE1, SNAIL, SKIL* and *ZEB1* and increase the expression of *ATM, BLM, FANCF, NBN, PMS1, RAD50* and *RDM1*.

As an orthogonal approach, we inhibited TGF-β mediated signaling by silencing *SMAD3*. As shown in Figure 8, TGF-β failed to increase the expression of *SERPINE1* and *SKIL* and to down-regulate the expression of *BLM, BRCA2, FANCF, NBN* and *RDM1* in cells with decreased *SMAD3* expression due to siRNA mediated knockdown.

Author response image 1.mRNA expression analysis of *SMAD3, SERPINE1, SKIL* and the indicated HDR genes in vehicle or TGF-β-treated cells relative to vehicle control upon siRNA mediated *SMAD3* knock down.si Scramble was used as an experimental control. mRNA expression was quantified by SYBR-green-based RT-qPCR. Each bar represents mean ± SD of 3 replicates from 2 independent experiments. (p-value * < 0.05, **< 0.005 paired t-test).**DOI:**
http://dx.doi.org/10.7554/eLife.21615.034

*The DR-GFP reporter system is used in U2OS cells (Figure 4 and supplementary data). These cells are not used in other experiments with major findings in this manuscript, are from mesenchymal origin, whereas the others are carcinomas, and have impaired TGF-b signaling. It would provide confidence to have this assay done in H1650 cells.*

As shown in Figure 4—figure supplement 2, when we performed a similar experiment in H1650 cells we observed that treatment of these cells with TGF-β decreased the efficiency of HDR repair mechanisms.

*Additional comments to be addressed are:*

The title and/or Abstract should provide a clear indication of the biological system under investigation (i.e., species name or broader taxonomic group, if appropriate). Please revise your title and/or Abstract with this advice in mind.

The Abstract has been modified to follow this guideline. Unfortunately, we struggle to modify the title due to the limitation in the number of characters. Any suggestions are welcome.

In all figures, please explain the error bars – what do they represent and if variance, how many experiments. Are the ones on the data from two experiments a range?

Additional details of the experiments and analyses have been added throughout the manuscript.

Throughout the paper where CNAs are discussed, please do not refer to "DNA breaks" or "break points". What is being quantifying is novel DNA junctions or join points. "Breaks" implies a mechanism of genome rearrangement that the authors do not know has occurred-one using DNA breakage. Models like microhomology-mediated replicative template switching and MMBIR are dominating modern understanding of CNA formation, and these are not break and join models. I suggest adopting either "join points" or new "DNA junctions", defining it for readers, and using throughout in all.

We appreciated this comment and have adopted the term “DNA join points” to describe what we previously referred to as “DNA breaks” or “DNA break points”.